# Quantum pure noise-induced transitions:
# A truly nonclassical limit cycle sensitive to number parity

Andy Chia[1], Wai-Keong (Dariel) Mok[2], Changsuk Noh[3] and Leong-Chuan Kwek[1,4,5,6]

**1** Centre for Quantum Technologies, National University of Singapore, Singapore
**2** Department of Physics, National University of Singapore, Singapore
**3** Department of Physics, Kyungpook National University, Daegu, South Korea
**4** National Institute of Education, Nanyang Technological University, Singapore
**5** Quantum Science and Engineering Centre, Nanyang Technological University, Singapore
**6** MajuLab, CNRS-UNS-NUS-NTU International Joint Research Unit, Singapore

## Abstract

It is universally accepted that noise may bring order to complex nonequilibrium systems. Most strikingly, entirely new states not seen in the noiseless system can be induced purely by including multiplicative noise—an effect known as pure noise-induced transitions. It was first observed in superfluids in the 1980s. Recent results in complex nonequilibrium systems have also shown how new collective states emerge from such pure noise-induced transitions, such as the foraging behavior of insect colonies, and schooling in fish. Here we report such effects of noise in a quantum-mechanical system without a classical limit. We use a minimal model of a nonlinearly damped oscillator in a fluctuating environment that is analytically tractable, and whose microscopic physics can be understood. When multiplicative environmental noise is included, the system is seen to transition to a limit-cycle state. The noise-induced quantum limit cycle also exhibits other genuinely nonclassical traits, such as Wigner negativity and number-parity sensitive circulation in phase space. Such quantum limit cycles are also conservative. These properties are in stark contrast to those of a widely used limit cycle in the literature, which is dissipative and loses all Wigner negativity. Our results establish the existence of a pure noise-induced transition that is nonclassical and unique to open quantum systems. They illustrate a fundamental difference between quantum and classical noise.

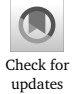

# 1 Introduction

Understanding the influence of noise on nonequilibrium dynamical systems is indispensable to several scientific endeavors, from climate science [1–6] to biological processes [7–14], from ecosystems and population dynamics [15–21] to their emergent behavior [22–26]. Knowledge of the interplay between noise and deterministic elements in a nonequilibrium system provides invaluable insight into many complex processes. To capture this interaction between noise and deterministic motion, scientists often use random processes that are system dependent, called multiplicative noise. Examples abound [27–33], dating back to more than a half century when Kubo first applied multiplicative noise to a linear system [34]. A simple consequence of

multiplicative noise which is also of contemporary interest is effective drifts (also known as noise-induced drifts) [35]. New work in complex systems has also shed light on the role of multiplicative noise in collective phenomena (see Ref. [36] and the references therein).

A recurrent message from the past several decades of research on nonlinear nonequilibrium systems is that noise can bring about states with structure, or order [30, 36–40]. The most astonishing feature of multiplicative noise in nonlinear nonequilibrium systems is their ability to induce structured states, or "phases," which without noise are completely absent. The occurrence of this phenomenon in macroscopic systems subject to a fluctuating environment has been designated a *pure* noise-induced (phase) transition by Horsthemke and Lefever (see Fig. 1), who pioneered the field known simply as noise-induced transitions [41–44].[1] However, not all multiplicative noise lead to such a drastic effect. In general, multiplicative noise might only induce a transition to a state that merely modifies a preexisting property in the system. An example of particular interest for this less dramatic type of noise-induced transition is a classical system with a limit cycle. In these systems, multiplicative noise shifts the Hopf bifurcation point (see Refs. [48–51] and similar results cited within). We shall see later in this work that multiplicative noise can have a more profound effect than just shifting the Hopf bifurcation point if the system is inherently quantum mechanical. That is, we will show how noise in quantum mechanics can induce a limit cycle in a system without one, thus achieving a pure noise-induced transition in contrast to its analogous classical system.

Pure noise-induced transitions were first observed in experiments using superfluid turbulence in the 1980s [53,54]. A more modern example appears in chemistry, where a pure noise-induced transition was shown to explain important properties of an enzymatic reaction [55]. New developments in collective dynamics have extended pure noise-induced transitions to systems of finite size. Internal noise in such systems can lead to multiplicative noise in collective variables which indicate emergent behavior [36, 56, 57]. This has been demonstrated in a model of foraging colonies with two food sources [56]. If the population falls below some threshold, the foragers change from being "undecided" to collective alternation between the two food sources [56]. Very recently, a pure noise-induced transition was also shown to explain schooling in fish [57].

In this work we show that pure noise-induced transitions can also occur in a microscopic system where quantum effects are essential. Our system is a nonlinearly damped oscillator coupled to a heat bath at temperature $T$, previously explored in the context of a linear system [58]. For $T > 0$, thermal fluctuations entering the oscillator can induce it to transition to a limit-cycle state which is otherwise absent at $T = 0$. When $T > 0$, our system resembles a Stuart–Landau oscillator on average. The transition to a limit cycle from the $T = 0$ case, may then be seen as a pure noise-induced Hopf bifurcation (the $T = 0$ case being akin to the noiseless system in the classical theory). Our focus on a system in a fluctuating environment mirrors the original theory of Horsthemke and Lefever [44]. This is especially suited to quantum systems as they are highly susceptible to environmental noise [59,60].

The noise-induced limit cycle of the microscopic oscillator has truly nonclassical traits. The transition to a limit-cycle state is shown to be unique to the microscopic oscillator, i.e. an analogous macroscopic oscillator subjected to multiplicative noise cannot be induced to undergo limit-cycle oscillations whatsoever. Moreover, the noise-induced limit cycle can sustain negative Wigner functions. This is in contrast to the conventional quantum Stuart–Landau oscillator in the literature for which Wigner negativity is always lost [61,62]. In phase space, the rotational flow of the microscopic oscillator is seen to have both classical and quantum contributions. The classical part can be explained by the macroscopic analog, but the quantum

---

[1]To avert a potential confusion, we mention that a different kind of noise-induced phase transitions has also been proposed [45], which also covers pure noise-induced phase transitions [46]. This is beyond the scope of our work but the interested reader should consult Ref. [47] for a discussion of the differences.

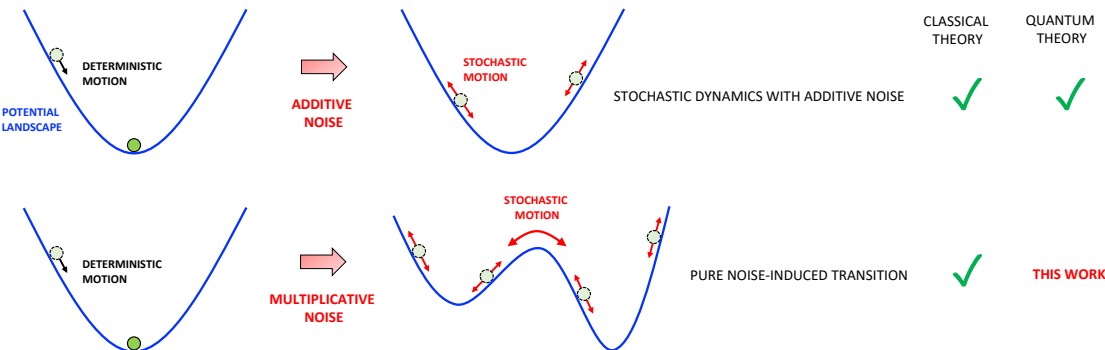

Figure 1: A potential-landscape picture [52] showing the difference between a system with additive noise and one that experiences a pure noise-induced transition with multiplicative noise. In the top row we depict a simple deterministic system with only one stable state (represented by the green ball at the bottom of the potential landscape). Upon introducing additive noise into the system, the motion of the system becomes stochastic. In this example, additive noise simply pushes the system around its deterministic fixed point at the bottom of the potential well. Depending on the realization of the noise, the system may be pushed in either direction with different strengths. We depict such stochastic motion using two-way arrows in red. Additive noise has been studied in both classical and quantum systems as discussed in the main text (also indicated here by the two green ticks on the right). In the bottom row we depict a pure noise-induced transition which relies on multiplicative noise. Multiplicative noise can induce completely new states which are absent in the noiseless system. This is shown here by the creation of a new double-well potential which now has two stable states. In this case the noise may even push the system over the middle barrier so that each stable state is only occupied for a finite time. Note that a pure noise-induced transition refers to the change in the steady-state behavior when a noiseless system becomes stochastic (here going from the single-well to double-well potential). Our paper studies how a pure noise-induced transition occurs in a quantum system, which to the best of our knowledge has not been considered up to now.

contributions are without any such classical roots. We then find the underlying probability flux responsible for the noise-induced limit cycle to be purely conservative, again in stark contrast to the conventional quantum Stuart–Landau model which is driven by a dissipative probability current. One may find a preview of our results in Table 1, which of course will not be referred to till later.

There has also been some recent results on noise-induced oscillations and transitions, including quantum systems, but in a different sense [63–66]. These results, which we shall discuss now, do not qualify as pure noise-induced transitions because they investigate phenomena whereby additive noise (which are system independent) induces the system to jump in and out of preexisting states. We illustrate this in Fig. 2 with a bistable system. Well known examples from classical theory are coherence resonance [67–75] and stochastic resonance [76–80]. These effects have in common the mechanism of noise-activated escape [81, 82]. The role of noise is simply to destabilize a system around a local basin of attraction, occasionally providing enough energy for the system to escape (if the basin is not globally attracting). In the case of coherence resonance, pulses resembling limit-cycle oscillations are produced using noise-activated escape in a system whose deterministic dynamics already contains a limit cycle (an excitable system [83]). Such pulsations have also come under the name of noise-

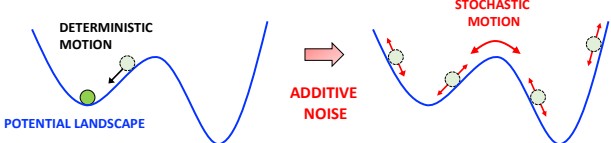

Figure 2: An additive-noise effect that has often been labelled as noise-induced bistability, but is not a pure noise-induced transition. It is important to differentiate this case from the pure noise-induced transition shown in the bottom row of Fig. 1. In the illustration here, the bistability of the system is already present prior to the introduction of noise in the sense that two stable states are available to be occupied. Because additive noise is independent of the system, all it can do is kick the system back and forth between the two wells. In the language of noise-induced transitions as defined by Horsthemke and Lefever [44], no new states have been induced by the addition of noise that is not already in the deterministic system. The change from deterministic to stochastic behavior illustrated here is therefore not a pure noise-induced transition.

induced or stochastic limit cycles [67, 68, 84, 85], and was recently studied in open quantum systems [63, 64]. Some authors have in fact defined a noise-induced "limit cycle" by using only a local basin of attraction and the destabilizing effect of additive noise [65]. For example, a deterministic system with a spiral sink, such as a damped harmonic oscillator, is said to have a noise-induced "limit cycle" in the presence of additive noise under Ref. [65].

For systems with two basins of attraction, additive noise with sufficient intensity may kick the system back and forth between the two basins. Such a state change has been called a noise-induced "transition" in the sense that noise activates a transition between two preexisting states, e.g. in Refs. [52, 86] (see Fig. 2). This is essentially what enables many noisy bistable phenomena, such as stochastic resonance.[2] This is also the case with recent work showing that additive noise can drive bistability between two synchronisation states in an open quantum system [66]. Thus, it is important to distinguish noise-activated bistability (illustrated in Fig. 2) from the pure noise-induced bistability in, e.g. Refs. [53, 54, 56], a point which Ref. [56] has highlighted and illustrated in Fig. 1. The effect of multiplicative noise on a bistable quantum system has been studied, but only with classical noise, and with the system in its classical limit [87].

We highlight again that our quantum limit cycle in this paper is the result of a pure noise-induced transition, and to achieve such dynamics, multiplicative noise is necessary. One way to see why Refs. [65]– [86] fall short of a pure noise-induced transition is to appeal to potential landscapes. If we associate the landscape with the system state, then additive noise simply pushes a system around on a preexisting landscape, but multiplicative noise can play a direct role in creating the potential landscape. We will explain again what makes our noise-induced transition pure more precisely using equations next, when we present our quantum model.

---

[2]Note that in stochastic resonance a deterministic sinusoidal force is also included in addition to a random force. However, the sinusoidal force only changes the height of the middle barrier relative to the bottom of the two wells in Fig. 2 in time. This effectively allows the likelihood of a jump occurring between the two wells in Fig. 2 to be modulated in time. It does not alter the basic character of additive noise, namely that it simply provides a random force which can only push the system around on the potential landscape.

## 2  Models

Our nonlinear oscillator is represented by a single bosonic degree of freedom, described by $\hat{a}$ and $\hat{a}^\dagger$ satisfying $[\hat{a}, \hat{a}^\dagger] = \hat{1}$. External multiplicative noise in $\hat{a}$ arises when two excitations are exchanged at a time with a heat bath. We interpret the system excitations to be photons, at frequency $\omega_0$, and the bath to be an ensemble of two-level atoms at temperature $T$. Exactly this type of nonlinear interaction had been of interest in quantum optics, especially in the context of two-photon absorption [88–97]. However, this body of literature makes no connection to multiplicative noise. The time-dependent state of the oscillator $\rho(t)$, may then be described by a Markovian master equation under standard approximations given by $d\rho(t)/dt = \mathcal{L}_\Uparrow \rho(t)$ where [59, 98–100]

$$\mathcal{L}_\Uparrow = -i\,\omega_0\,[\hat{a}^\dagger\hat{a}, \cdot\,] + \kappa_\Downarrow\,\mathcal{D}[\hat{a}^2] + \kappa_\Uparrow\,\mathcal{D}[\hat{a}^{\dagger 2}]. \tag{1}$$

Note the dot denotes the position of $\rho(t)$ when acted upon by $\mathcal{L}_\Uparrow$. The parameters $\kappa_\Downarrow$ and $\kappa_\Uparrow$ are positive real numbers, proportional to the atomic ground-state and excited-state populations respectively. We have also defined, $\mathcal{D}[\hat{c}] = \hat{c} \cdot \hat{c}^\dagger - (\hat{c}^\dagger\hat{c}\cdot + \cdot\hat{c}^\dagger\hat{c})/2$ for any $\hat{c}$. We refer to $\mathcal{D}[\hat{c}]$ as a dissipator, and $\hat{c}$ a Lindblad operator. It captures the bath's influence on the system. When $T \longrightarrow 0$, all the atoms occupy their ground state and we find $\kappa_\Uparrow \longrightarrow 0$. Thus any new physics arising from a nonzero $\kappa_\Uparrow$ must be attributed to thermal noise.

The role of (1) is akin to the Fokker–Planck equation in the macroscopic theory of noise-induced transitions [44]. And likewise, the steady state of (1) will be of central importance to us. It has been derived by noting that (1) conserves photon-number parity, i.e. $d\langle(-1)^{\hat{n}}\rangle/dt = 0$, where $\hat{n} = \hat{a}^\dagger\hat{a}$ [90, 93]. Here we express it as,

$$\rho_{ss} = \wp_+ \rho_+ + \wp_- \rho_-, \tag{2}$$

where $\wp_+$, $\wp_-$ are respectively the probability for $\rho(0)$ to be in an even or odd parity subspace, and $\rho_+$, $\rho_-$ are, for $K = \kappa_\Uparrow/\kappa_\Downarrow$,

$$\rho_+ = (1-K)\sum_{n=0}^{\infty} K^n \, |2n\rangle\langle 2n|, \tag{3}$$

$$\rho_- = (1-K)\sum_{n=0}^{\infty} K^n \, |2n+1\rangle\langle 2n+1|. \tag{4}$$

When the noise is explicitly taken into account by using stochastic equations (see below), a microscopic understanding of (1) in terms of elementary atom-photon interactions can be reached. We summarise this in Fig. 3(a). Most notably, we find singly-stimulated emissions in the $\hat{a}^{\dagger 2}$ dissipator [58, 89]. This is vital to understanding how a limit cycle is physically possible in a model with only two-photon processes. In a model with two-photon loss, conventional wisdom requires that it be supplemented by one-photon gain if a stable limit cycle is to arise. The conventional model for a quantum limit cycle which has been widely used in the literature is thus $d\rho(t)/dt = \mathcal{L}_\uparrow \rho(t)$ [61, 62], where

$$\mathcal{L}_\uparrow = -i\,\omega_0\,[\hat{a}^\dagger\hat{a}, \cdot\,] + \kappa_\Downarrow\,\mathcal{D}[\hat{a}^2] + \kappa_\uparrow\,\mathcal{D}[\hat{a}^\dagger]. \tag{5}$$

In contrast to (1), the one-photon gain is introduced as an independent process using a second bath (which can be thought of as a perfectly inverted atomic medium). The singly-stimulated emission in Fig. 3(a) has a similar effect as the $\hat{a}^\dagger$ dissipator in (5). This is the first telltale sign that thermal noise from the two-photon dissipator can induce a quantum limit cycle which is otherwise absent at $T = 0$. As we shall see, the different physics in $\mathcal{L}_\Uparrow$ and $\mathcal{L}_\uparrow$ lead to limit cycles with very different characteristics.

The same approximations leading to (1) also gives a quantum stochastic differential equation in Stratonovich form [58],

$$d\hat{a}(t) = \left[-i\,\omega_0\,\hat{a}(t) - (\kappa_\Downarrow - \kappa_\Uparrow)\,\hat{a}^\dagger(t)\,\hat{a}^2(t)\right]dt + \hat{a}^\dagger(t) \circ d\hat{W}(t). \tag{6}$$

External thermal fluctuations are now explicit in (6), represented by a quantum Wiener increment $d\hat{W}(t)$ (a bath operator) [101, 102]. Stratonovich equations are denoted by a circle in system-noise products [44, 103, 104]. The effect of noise can be made clear by either calculating the average of (6), or by converting (6) to its Itô form. Taking the latter approach, we find the Itô equation

$$d\hat{a}(t) = \left[-i\,\omega_0\,\hat{a}(t) + 2\kappa_\Uparrow\,\hat{a}(t) - (\kappa_\Downarrow - \kappa_\Uparrow)\,\hat{a}^\dagger(t)\,\hat{a}^2(t)\right]dt + \hat{a}^\dagger(t)\,d\hat{W}(t). \tag{7}$$

An inspection of (6) and (7) gives the expectation value $\langle \hat{a}^\dagger(t) \circ d\hat{W}(t)\rangle = 2\kappa_\Uparrow \langle \hat{a}(t)\rangle$. The $2\kappa_\Uparrow \hat{a}(t)$ in (7) is thus a quantum noise-induced drift. Noise-induced drifts are a well-known concept in classical statistical physics [35, 105], which has just begun to be explored for quantum systems [58, 106].[3] The microscopic interpretation of (6) is shown in Fig. 3(b) [58]. With (7) at hand, we can now explain what makes our noise-induced transition pure (with the actual transition being presented in Sec. 3). We can minimize the external thermal noise entering the system by setting the bath temperature to zero. Then $\kappa_\Uparrow = 0$, and the system reduces to a nonlinearly damped oscillator. In this case one would not even expect a limit cycle from (7) because without the linear gain given by $2\kappa_\Uparrow\hat{a}$, there is no counterbalance to the nonlinear damping given by $-(\kappa_\Downarrow - \kappa_\Uparrow)\hat{a}^\dagger\hat{a}^2$, and no reason why such a system should stabilize to some finite amplitude in the long-time limit. This highlights a fundamental difference between multiplicative noise and additive noise. Multiplicative white noise can induce additional terms on the order of $dt$, which may then generate a new potential landscape for the system. Additive noise on the other hand cannot introduce such new processes on the order of $dt$, and hence cannot produce the kind of drastic modifications which multiplicative noise is capable of.

Particularly important for us is the quantum (i.e. non-commutative) nature of noise, as expressed by the quantum Itô rules for ideal white noise [107–110],

$$d\hat{W}(t)\,d\hat{W}^\dagger(t) = 4\kappa_\Downarrow\,dt, \quad d\hat{W}^\dagger(t)\,d\hat{W}(t) = 4\kappa_\Uparrow\,dt. \tag{8}$$

When $T \longrightarrow \infty$, the atoms occupy their excited and ground states equally so $\kappa_\Downarrow - \kappa_\Uparrow \longrightarrow 0$, and the noise becomes classical (i.e. commuting). The nonlinearity in (6) and (7) vanishes in this limit. In the opposite limit of $T \longrightarrow 0$, the noise becomes most quantum, but the noise-induced drift vanishes and the distinction between (6) and (7) disappears. Hence, the thermal noise cannot be entirely quantum nor entirely classical to induce a limit cycle. This paper tackles the full nonlinear problem for which $0 < T < \infty$ and $0 < \kappa_\Uparrow < \kappa_\Downarrow$. A pure noise-induced transition occurs from $\kappa_\Uparrow = 0$ (no external noise) to $\kappa_\Uparrow > 0$ (some external noise).

It will also be useful to compare our quantum model to a classical model which emulates the quantum dynamics as closely as possible. We thus propose the following macroscopic model based on (6),

$$d\alpha(t) = \left[-i\,\omega_0\,\alpha(t) - \Delta\,|\alpha(t)|^2\,\alpha(t)\right]dt + \alpha^*(t) \circ dW(t), \tag{9}$$

where $\Delta > 0$. We can then show that a noise-induced drift arises as in the quantum case. The Itô equivalent of (9) given by

$$d\alpha(t) = \left[-i\,\omega_0\,\alpha(t) + 2\kappa\,\alpha(t) - \Delta\,|\alpha(t)|^2\,\alpha(t)\right]dt + \alpha^*(t)\,dW(t), \tag{10}$$

---

[3]See Sec. 4 of Ref. [35], especially the bullet point "quantum noise-induced drift." Our results are also closely related to the bullet points on "effects of multiplicative noise on steady-state distributions," and "noise-induced bifurcations," but for a quantum system.

(a)

$$\frac{d}{dt}\rho(t) = \kappa_\Downarrow \, \mathcal{D}[\hat{a}^2]\rho(t) + \kappa_\Uparrow \, \mathcal{D}[\hat{a}^{\dagger 2}]\rho(t)$$

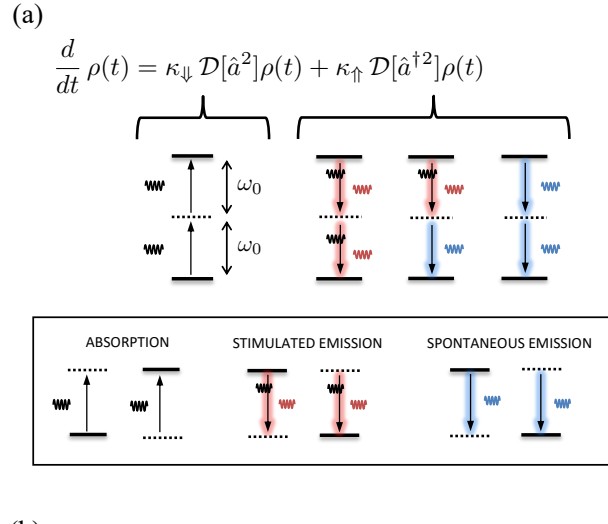

(b)

$$d\hat{a}(t) = -(\kappa_\Downarrow - \kappa_\Uparrow)\,\hat{a}^\dagger(t)\,\hat{a}^2(t)\,dt + \hat{a}^\dagger(t)\circ d\hat{W}(t)$$

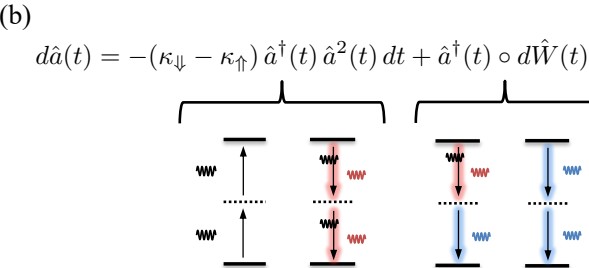

Figure 3: Microscopic view of multiplicative quantum noise in a rotating frame with frequency $\omega_0$. Solid lines represent the atomic excited and ground states, while dotted lines denote virtual states (states with vanishing lifetime). Multiplicative noise induces linear amplification (noise-induced drift). At the microscopic level, noise comes from spontaneous emission while amplification comes from stimulated emission. The multiplicative noise is thus a two-photon emission where one of the photons is stimulated [58]. (a) The quantum master equation (1) interpreted as two-photon processes (top), and a graphical key for reading the atom-photon interactions (bottom). Note the noise-induced drift is embedded in $\mathcal{D}[\hat{a}^{\dagger 2}]$. This is the underlying mechanism supporting the quantum and pure noise-induced limit cycle in this paper. (b) The Stratonovich quantum stochastic differential equation in terms of two-photon processes.

except now $dW(t)$ is a complex Wiener increment satisfying

$$dW^*(t)\,dW(t) = 4\kappa\,dt\,. \tag{11}$$

We will find that no noise-induced transition to a limit cycle is possible in this model whatsoever (i.e. for any value of $\kappa$ and $\Delta$).

Although our classical model is designed to mimick the quantum system as closely as possible, there are nevertheless some intrinsic differences. First, one often extracts the macroscopic limit of an open quantum system by performing a system-size expansion [98]. However, this method is only capable of extracting macroscopic systems with additive noise [61, 98]. Thus it cannot be employed to investigate whether multiplicative noise can induce limit cycles in a classical system. The system-size expansion thus provides the rigorous basis for the lack of a semiclassical limit (also referred to as the macroscopic limit) in our quantum oscillator. In this regard, (9) and (10) should be thought of as commutative versions of (6) and (7) respectively, and not actually their macroscopic limits, because such limits do not exist. Our use of

the word "macroscopic" as a descriptor for the classical model is therefore qualified. Second, thermal noise entering the microscopic oscillator as defined in (8) will have vacuum fluctuations, which is a quantum property. That is, the bath cannot be made to produce exactly zero noise even when all atoms are in their ground state. This can also be understood from the necessity to preserve the canonical commutation relation for the system at all times and at all temperatures [58]. If on the other hand we set $\kappa = 0$ in the classical stochastic equations, they become completely deterministic.

## 3 Noise-induced transitions

### 3.1 Preliminaries and the $T = 0$ case

To demonstrate noise-induced transitions in quantum systems we follow a similar approach as the classical theory [44], whereby the transition is characterized by the mode of the system's steady-state probability density. This is formally known as phenomenological bifurcations, or P-bifurcations for short [111–113]. Essentially the same idea applies for an open microscopic system, but to their quasiprobability distributions. The idea had been noted early on in quantum optics [114, 115]. Its use is now prevalent in physics (often without reference to P-bifurcations), such as in defining limit cycles near a Hopf bifurcation [61, 62], relaxation oscillations [116], amplitude and oscillation death [117–120], and Turing instabilities [121]. It is important to note here that P-bifurcations refer to the entire class of bifurcations inferred by looking at qualitative changes in the steady-state distribution as some system parameter is changed. It can therefore also be taken as a method for identifying nonlinear features and bifurcations, and should not be confused with the Hopf bifurcation which is the focus of this paper.[4]

Note the steady state for $T = 0$ cannot be obtained from setting $\kappa = 0$ in (2)–(4), so it is worthwhile to quickly review this case. In the absence of thermal fluctuations the system reduces to a nonlinearly damped oscillator without limit cycles. In this case the steady state is constrained to a two-dimensional subspace, spanned by the Fock states $|0\rangle$ and $|1\rangle$. This is because $\mathcal{D}[\hat{a}^2]$ damps all even Fock states to $|0\rangle$, and all odd Fock states to $|1\rangle$ [90]. All coherences between $|2n\rangle$ and $|2n+1\rangle$ also ultimately contribute to the coherence between $|0\rangle$ and $|1\rangle$ [93]. In fact, for $\kappa_\Uparrow = 0$, exact expressions for $\rho(t)$ in the number basis can be derived for all $t$ [90, 93]. Much later an operator form of $\rho(t)$ was obtained [122]. Interestingly, the problem was revisited again from symmetry and conservation considerations [123]. We will come back to this briefly in Sec. 4.1. For now it suffices to say that it expresses the steady-state solution in terms of conserved quantities. From this perspective, the $\kappa_\Uparrow = 0$ case differs from the $\kappa_\Uparrow > 0$ in that an extra conserved quantity related to coherences may be found in addition to parity (Appendix E contains more details but is not necessary until Sec. 4.1, where we actually discuss ideas relevant to conservation and symmetry principles).

Our central result here is that limit-cycle states arise as a consequence of adding thermal noise to the damped oscillator and that they are a fundamentally new set of states absent in the noiseless system. Such steady states are interesting because they are stable-amplitude oscillations which are only possible with nonlinear dynamics. That is, we do not consider the steady state of a dynamical system defined only by simple-harmonic motion to have a limit cycle even if its Wigner function may look like one.

---

[4]An alternative way to define bifurcations in noisy systems is via dynamical bifurcations, or D-bifurcations for short [113]. However, this method is mathematically much more demanding and nontrivial to generalize to quantum theory. This invariably makes P-bifurcations the preferred method for classifying bifurcations and nonlinear features amongst physicists.

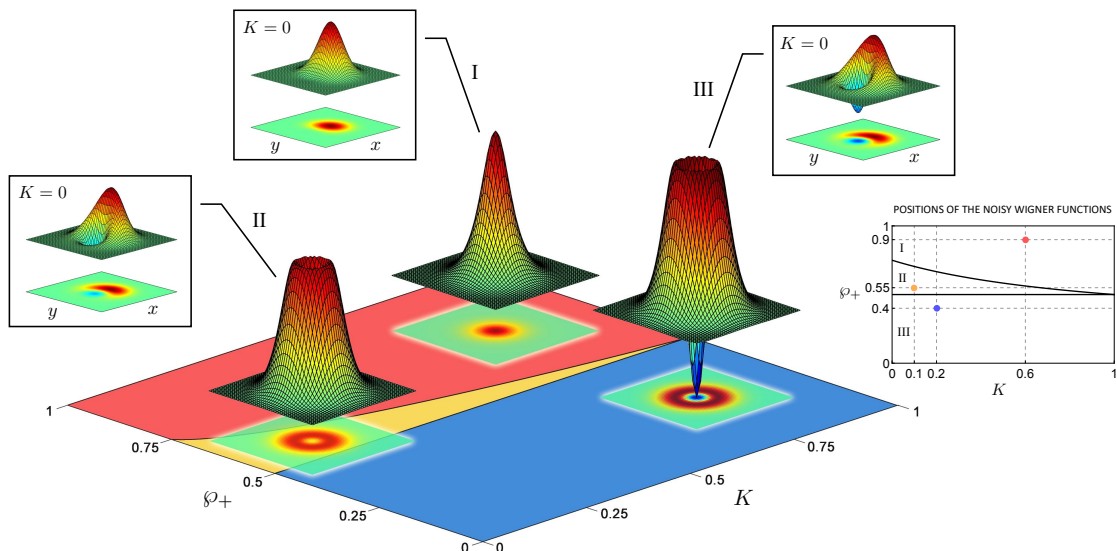

Figure 4: Phase diagram depicting the steady-state Wigner functions of (1) for different values of $(K, \wp_+)$ on a unit square characterized by the three phases explained in the text. The $(K, \wp_+)$ coordinates of the $K > 0$ Wigner functions are indicated in an inset on the right. The contour plots of each Wigner function are also displayed. For each phase, we have chosen a $\rho(0)$ (not shown) such that the noiseless steady states are all pure, parameterized by $|\psi_{ss}\rangle = \sqrt{\wp_+}|0\rangle + \sqrt{\wp_-}|1\rangle$. The noisy steady states correspond to the same $\rho(0)$ in each phase. We have also scaled the Wigner functions arbitrarily for ease of visualization. Phase I (red region): No noise-induced transition occurs. The noisy Wigner function has $(K, \wp_+) = (0.6, 0.9)$ and represents a stable origin under P-bifurcations. The noiseless Wigner function in the inset has $(K, \wp_+) = (0, 0.9)$. Phase II (yellow region): A pure noise-induced transition defined by $K = 0 \longrightarrow K = 0.1$ along $\wp_+ = 0.55$. The noisy Wigner function is always positive and attains the characteristic craterlike form for Stuart–Landau oscillators. Phase III (blue region): A pure noise-induced transition defined by $K = 0 \longrightarrow K = 0.2$ along $\wp_+ = 0.4$. The shape of the noisy Wigner function remains craterlike, but now the bottom of the crater can go below zero.

## 3.2 Pure noise-induced limit cycle ($T > 0$)

### 3.2.1 Classification of transitions

The steady-state Wigner distribution for the noisy $\rho_{ss}$ can in fact be derived in closed form. This is of great value for characterizing how the system behaves as a function of the external noise. Nontrivial open quantum systems with an exactly solvable steady-state quasiprobability distribution are few and far between. Indeed, the exact solvability had also played an important role in the macroscopic theory of noise-induced transitions and stochastic bifurcations. It is why one-variable systems were initially preferred by Horsthemke and Lefever [44, 112]. The Wigner function of (2) for $0 < K < 1$ in Cartesian coordinates is given by (see Appendix A),

$$W_{ss}(x, y) = \wp_+ W_+(x, y) + \wp_- W_-(x, y), \tag{12}$$

where $x$ and $y$ also correspond to the quadratures of the quantum nonlinear oscillator, and

$$W_+(x,y) = \gamma\, e^{-(x^2+y^2)/2}\left[\frac{e^{-\eta(x^2+y^2)}}{1-\sqrt{K}} + \frac{e^{\lambda(x^2+y^2)}}{1+\sqrt{K}}\right], \tag{13}$$

$$W_-(x,y) = \frac{\gamma}{\sqrt{K}}\, e^{-(x^2+y^2)/2}\left[\frac{e^{\lambda(x^2+y^2)}}{1+\sqrt{K}} - \frac{e^{-\eta(x^2+y^2)}}{1-\sqrt{K}}\right]. \tag{14}$$

For ease of writing we have defined the constants (all positive),

$$\gamma = \frac{1-K}{4\pi}, \quad \eta = \frac{\sqrt{K}}{1-\sqrt{K}}, \quad \lambda = \frac{\sqrt{K}}{1+\sqrt{K}}. \tag{15}$$

Note that since $\wp_- = 1-\wp_+$, $W_{\text{ss}}$ can be parameterized by $(K, \wp_+)$ on a unit square. In addition, since $W_{\text{ss}}$ is a function of only $x^2 + y^2$, no information is lost by working instead with the single-variable function $W(r) \equiv 4W_{\text{ss}}(2r\cos\phi, 2r\sin\phi)$ (not to be confused with a change of measure to polar coordinates). Then by P-bifurcations, the radius of a limit cycle, when it exists, is given by its mode, $r_\star \equiv \arg\max W(r)$. The nonclassical nature of our oscillator means that its Wigner function can also be negative. For the specific $W(r)$ defined above, we find $W(r) < 0$ if and only if $W(0) < 0$ (see Appendix B). We then find on the unit square parameterized by $(K, \wp_+)$, all steady states to fall under one of the three phases defined by its $r_\star$ and $W(0)$ (see Appendix B). We label these phases I, II, and III, which are defined as follows

I : A stable origin corresponding to $r_\star = 0$, $W(0) > 0$. Occurs when $\wp_+ > (3+K)/4(1+K)$.

II : Limit cycle with a positive Wigner function corresponding to $r_\star > 0$, $W(0) > 0$. Occurs when $1/2 < \wp_+ < (3+K)/4(1+K)$.

III : Limit cycle with a negative Wigner function corresponding to $r_\star > 0$, $W(0) < 0$. Occurs when $\wp_+ < 1/2$.

We illustrate the qualitative behavior of $W_{\text{ss}}$ for each phase in Fig. 4. Note the Wigner functions have been scaled and do not appear according to their $(K, \wp_+)$ coordinates in order to make the three-dimensional illustration feasible (see Fig. 4 caption for parameter values and other information). A more detailed discussion of the parameter space is provided in Fig. 5.

To capture the steady-state behavior over the entire parameter space we resort to $W(r)$. Ten samples of $W(r)$ are shown in Fig. 5, labeled 1 to 10. As can be seen in phase I, initial states with a sufficiently high population of even Fock states cannot be induced to undergo limit-cycle oscillations by adding noise (Fig. 4 and 1, 7, 8 in Fig. 5). Even Fock states have their Wigner-function modes at the origin, but not odd Fock states. Therefore for a given $K$, increasing the occupation of even Fock states in $\rho(0)$ tends to maximize $W(0)$, preventing the occurrence of a limit cycle ($2 \to 1$ or $8 \to 7$ in Fig. 5). Reducing the population of even Fock states then allows thermal fluctuations to induce limit-cycle behavior in the oscillator. In fact, we observe a supercritical Hopf bifurcation with respect to $\wp_+$ ($1 \to 2 \to 3$ in Fig. 5). This is a nonclassical trait as photon-number parity is nonexistent in classical physics. Although "number parity" is not a conventional bifurcation parameter, the limit-cycle size at birth does acquire the characteristic square-root dependence on parameters (here $\wp_+$ and $K$) for a supercritical Hopf bifurcation (see Appendix B).

In phase II, $W(r)$ (and hence also $W_{\text{ss}}$) is positive everywhere (3 and 9 in Fig. 5). Somewhat counter intuitively, the more noise the oscillator experiences, the more likely it is to be found near the origin (also $1 \to 7$ in phase I). Thus if the oscillator already has a noise-induced limit cycle (i.e. is in phase II), then adding more noise will destroy it ($3 \to 6 \to 8$ in Fig. 5). Such deleterious effects of multiplicative noise has also been shown for a classical limit cycle where
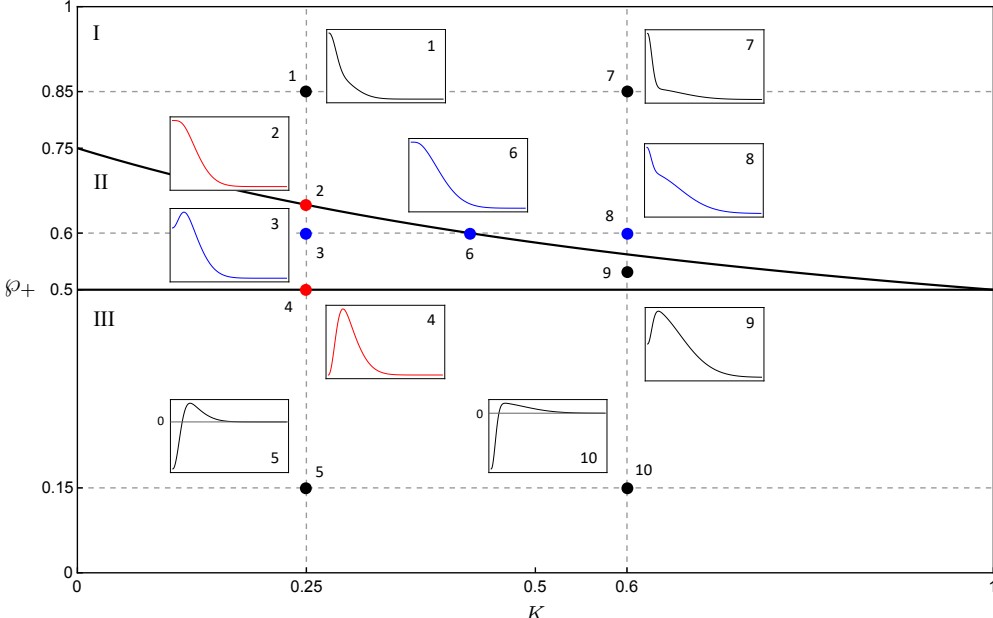

Figure 5: Behavior of $W(r)$ at ten different values of $(K, \wp_+)$, labeled from 1 to 10. Borderline cases are illustrated by points 2, 4, and 6. Note that point 6 is situated at $(0.43, 0.6)$ ($K$ value rounded to two decimal places), and point 9 is at $(0.6, 0.53)$. All insets show $W(r)$ from zero and above in the region $\wp_+ \geq 0.5$. For $W(r)$ in $\wp_+ < 0.5$ (i.e. points 5 and 10), $W = 0$ is marked on the vertical axis in the insets. A Hopf bifurcation occurs when I $\rightarrow$ II along $\wp_+$ ($1 \rightarrow 2 \rightarrow 3$), while an inverse stochastic bifurcation occurs for II $\rightarrow$ I along $K$ ($3 \rightarrow 6 \rightarrow 8$). A II $\rightarrow$ III crossover cannot happen without $W(0)$ becoming negative. We find for all $K$ values that $W(0)$ is controlled by $\wp_+$ as illustrated in the sequence of changes along $K = 0.25$ ($1 \rightarrow 2 \rightarrow 3 \rightarrow 4 \rightarrow 5$), or along $K = 0.6$ ($7 \rightarrow 8 \rightarrow 9 \rightarrow 10$). Phase III is "quantum protected" where the noise-induced oscillations are robust against thermal noise in retaining both its nonclassicality and limit-cycle behavior.

it has been referred to as an inverse stochastic bifurcation [51]. Here we see a quantum analog of this behavior.

Interestingly, the noise-induced transition becomes truly nonclassical if $\wp_+$ is further reduced to phase III ($3 \rightarrow 4 \rightarrow 5$ and $9 \rightarrow 10$ in Fig. 5). It can be shown that initial states with $\wp_+ < 1/2$ have negative Wigner functions (see Appendix B). However, thermal noise cannot destroy this Wigner negativity. Thus in phase III the noise-induced limit cycle remains nonclassical. This is in contrast with the conventional limit cycle obtained from $\mathcal{L}_\uparrow$ in (5), whose steady-state Wigner function is always positive. Moreover, adding more noise to the oscillator does not induce an inverse stochastic bifurcation, i.e. its limit cycle never vanishes and the limit cycle in this sense may be said to be "quantum protected" ($5 \rightarrow 10$ in Fig. 5). However, increasing the noise intensity does smear out the oscillator's distribution over phase space, producing a $W(r)$ with a long tail (see Appendix B for more on the tail behavior). It is also worth mentioning here that limit cycles with a negative Wigner function have been studied before in the context of optomechanics [124, 125]. However, the master equation for such systems are much more complicated than (1), and the limit cycles so obtained are not due to thermal noise.

Even if our pure noise-induced transition falls under phase II, it can still be regarded as nonclassical in that the macroscopic analog given by (9) and (10) cannot yield a limit cy-

cle whatsoever. This can be shown by considering its associated Fokker–Planck equation $\partial P(x, y, t)/\partial t \equiv \mathscr{L}P(x, y, t)$ and showing that the solution to $\mathscr{L}P_{ss}(x, y) = 0$ is given uniquely by (see Appendix C)

$$P_{ss}(x, y) = \frac{\Delta}{8\pi\kappa} e^{-\Delta(x^2+y^2)/8\kappa}. \tag{16}$$

This is a two-dimensional Gaussian centred at the origin corresponding to two independent processes $X(t)$ and $Y(t)$ with steady-state variances given by $\sigma_X^2 = \sigma_Y^2 = 4\kappa/\Delta$. Clearly $P_{ss}$ has a mode at the origin independently of the initial distribution and $(\Delta, \kappa)$.[5] Where possible, we include a comparison of different attributes between $\mathscr{L}$, $\mathcal{L}_\uparrow$, and $\mathcal{L}_\Uparrow$ in Table 1.

Previous work in noise-induced transitions on limit-cycle systems have only discovered noise-induced shifts of Hopf bifurcations, an effect belonging to the more typical class of noise-induced transitions [48–51] [take e.g. equation (12) of Ref. [51] and set $\mu = \sigma_2 = 0$]. Our reason for considering (9) and (10) is to show that even a classical model which best mimicks (6) and (7) do not show a pure noise-induced transition.

### 3.2.2 Dependence on photon-number parity

Since our steady states depend on the initial state via the photon-number parity, one might wonder if the $W_{ss}(x, y)$ seen in regions II and III indeed resemble limit cycles, as opposed to non-isolated orbits (such as the orbits of an undamped harmonic oscillator, or a simple Lotka–Volterra model). Here we affirm their interpretation as a generalization of classical limit cycles to quantum phase space, but some justification is required.

The problem with regarding the steady states in regions II and III as non-isolated orbits is that such an interpretation only makes sense for the most classical initial states—i.e. coherent states $|\alpha\rangle$. In this case we can in fact deduce a one-to-one correspondence between the mean photon number of the steady state $\langle\hat{n}\rangle_{ss}$, and that of the initial state $|\alpha|^2$, given by $\langle\hat{n}\rangle_{ss} = 2K/(1-K) + \exp(-|\alpha|^2)\sinh(|\alpha|^2)$ (see Appendix B and D). This result is akin to the non-isolated orbits of an energy-conserving system like the undamped harmonic oscillator, where an initial phase-space point of a given energy will only move in an orbit of the same energy (though here, $|\alpha|^2$ maps to a different energy given by $\langle\hat{n}\rangle_{ss}$, and we do not have energy conservation, but rather number parity). Naturally, this conformity to classical intuition tempts one to conclude that our quantum oscillator actually has non-isolated orbits instead of a limit cycle (which should be isolated). However, for limit cycles defined in phase space, as is the case here (in contrast to limit cycles in Hilbert space), the classical intuition just described fails for a general initial state. The Wigner function for an arbitrary initial state can be highly delocalised in quantum phase space,[6] and even nonclassical. For such general initial states the one-to-one correspondence between steady-state orbits and the initial state breaks down.

In fact, our closed orbits conform to the characteristic property of limit cycles in classical phase space, namely that at least one other trajectory spirals towards it [131]. We know already that $\wp_+$ divides all initial states into two subsets, one of which we may identify as a quantum limit cycle (regions II or III in Fig. 5). For a fixed $K$, any two initial states with the same $\wp_+$ in region II or region III will converge to the same closed orbit represented by

---

[5]An alternative way to make sense of the stability of the origin is through nonequilibrium potentials [52, 126, 127] (see also Chap. 7 in Vol. 1 of Ref. [28]). For our macroscopic oscillator this is given by the exponent of (16). One approach to constructing the nonequilibrium potential is via the so-called A-type stochastic differential equations [127–129]. We have used this technique and shown that it is consistent with P-bifurcations as desired (not presented in this paper but see Example 2 in Appendix B of Ref. [130] for a similar classical system to ours).

[6]Of course, one may argue that an ensemble of nonlinear classical systems initialized in different parts of phase space would also be delocalized. However, this is quite different to the quantum evolution of a Wigner function because each initial phase-space point in the classical ensemble evolves independently of every other phase-space point. The evolution of the Wigner function for a nonlinear quantum system on the other hand cannot be understood as the result of propagating independent initial conditions in quantum phase space.

the same $W_{ss}(x, y)$. Two such initial states can be made to have very different quasiprobability distributions in quantum phase space, e.g. corresponding to intial phase-space points concentrated in different areas of phase space.

Fundamentally, photon-number parity has no analog in classical dynamics as photons do not exist classically. The classical condition that states near an isolated orbit must be attracted to it, or repelled by it, cannot be sensibly imposed on a quantum oscillator. Trajectories in phase space are actually ill-defined in quantum systems. In this regard, our proposed quantum limit cycle is a generalization of the classical limit cycle to quantum phase space where the intrinsically quantum feature of parity conservation plays a central role.

### 3.3 Noise-induced nonclassicality

While the lack of a pure noise-induced transition in the classical model above already indicates that our pure quantum noise-induced transition is genuinely quantum, even in phase II, here we provide an analysis using alternative notions of nonclassicality. Interestingly, we are able to show that even a subset of states in phase I can be nonclassical. Physically however, this should not be too surprising, given the two-photon nature of our quantum model.

We start by considering the standard definition of nonclassicality from quantum optics. This defines an arbitrary density operator to be nonclassical if its Glauber–Sudarshan quasiprobability distribution fails to be a genuine probability distribution [132]. Using this, we can show that for $K > 0$, the quantum channel which maps $\rho(0)$ to $\rho_{ss}$ actually generates nonclassicality, not just preserves it. To do so we consider an initial vacuum state, i.e. $\rho(0) = |0\rangle\langle 0|$, which is a classical state. Due to parity conservation, it is mapped to $\rho_{ss} = \rho_+$, which obviously has $\langle n|\rho_{ss}|n\rangle = 0$ for any odd $n$ [recall (2)–(4)]. Then expanding $\rho_{ss}$ in terms of coherent states $|\alpha\rangle$ we have, for odd $n$,

$$\int_{\mathbb{C}} d^2\alpha\, \bar{P}_{ss}(\alpha, \alpha^*)|\langle n|\alpha\rangle|^2 = 0\,, \tag{17}$$

where $\bar{P}_{ss}(\alpha, \alpha^*)$ is the Glauber–Sudarshan quasiprobability distribution of $\rho_{ss} = \rho_+$. It then follows that (17) can be true only if $\bar{P}_{ss}(\alpha, \alpha^*)$ is not a proper probability distribution since $|\langle n|\alpha\rangle|^2$ is positive (see also footnote 2 of Ref. [133]). Thus the map $\rho(0) \longrightarrow \rho_{ss}$ generates nonclassicality. Note that for $K = 0$, i.e. without thermal noise, $\rho(0) = |0\rangle\langle 0| = \rho_{ss}$, so the steady state remains classical. Hence for $K > 0$, the nonclassicality in $\rho_{ss}$ may be aptly said to be noise induced.

In terms of the phase diagram in Fig. 5, the above argument shows that all steady states along the line $\wp_+ = 1$ are nonclassical, with the exception of the two end points $(K, \wp_+) = (0, 1)$ and $(K, \wp_+) = (1, 1)$ (remember that $K = 1$ only if the bath temperature is infinite). Since we have the exact form of the steady state, it should be possible to determine, at least in principle, the nonclassicality of more steady states on the phase diagram. To this end we consider the Mandel $Q$ parameter. This is defined in terms of the statistics of $\hat{n} = \hat{a}^\dagger \hat{a}$, by

$$Q = \frac{\langle \hat{n}^2 \rangle - \langle \hat{n} \rangle^2}{\langle \hat{n} \rangle} - 1\,. \tag{18}$$

If a given density operator leads to $Q < 0$, then it is guaranteed to be nonclassical. Note that like Wigner negativity, this is only a sufficient condition for nonclassicality, so the negativity of $Q$ is also only a witness. If $Q > 0$ then no conclusion about the nonclassicality of the state can be drawn. Physically, the Mandel $Q$ parameter captures nonclassicality by sub-Poissonian photon statisitics, when the photon-number variance becomes smaller than the mean. Note that $Q$ has a minimum value of $-1$, so the condition for nonclassicality may be stated as $-1 \leq Q < 0$.

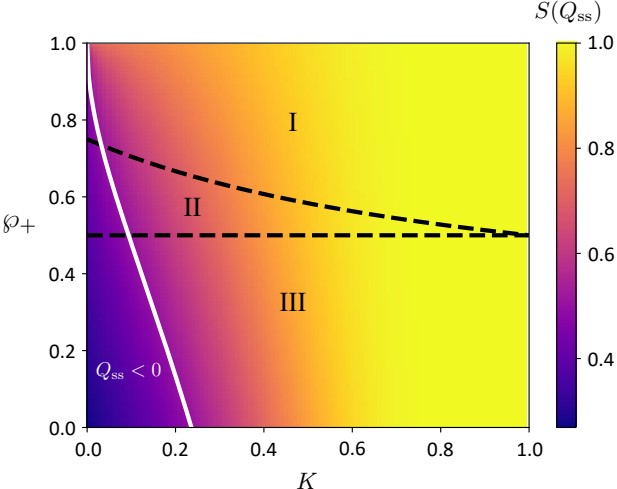

Figure 6: Contour plot of $S(Q_{ss})$ as a function of $K$ and $\wp_+$. The black dashed lines denote the boundaries for phases I, II and III. The white solid line corresponds to $Q_{ss} = 0$, and the region to the left of the white solid line have $Q_{ss} < 0$, which is guaranteed to be nonclassical.

Using expressions (2)–(4) for $\rho_{ss}$ in (18), we find

$$Q_{ss} = \frac{2}{1-K} + \frac{2K}{1+K-\wp_+(1-K)} + \wp_+ - 3 \, . \tag{19}$$

Note that if $(K, \wp_+) = (0, 0)$ then $Q_{ss} = -1$, saturating its lower bound. In this case $\rho_{ss} = |1\rangle\langle 1|$, which is clearly nonclassical. We can derive conditions for $K$ and $\wp_+$ in order for $Q < 0$. Some calculation gives

$$0 \le K < \frac{\sqrt{5 - 4\wp_+(2 - \wp_+)} - 3}{1 + \wp_+(2 - \wp_+)} + 1, \quad 0 \le \wp_+ < 1 \, . \tag{20}$$

The set of $(K, \wp_+)$ points satisfying (20) then prescribes a region in Fig. 5 corresponding to nonclassical steady states. We show this region in Fig. 6. Note however that $Q_{ss} \longrightarrow \infty$ when $K \longrightarrow 1$, so $Q_{ss} \in [-1, \infty)$. This range of $Q_{ss}$ makes it difficult to visualize so we map it to a finite interval using the sigmoid function, given by

$$S(Q_{ss}) = \left(1 + e^{-Q_{ss}}\right)^{-1} \, . \tag{21}$$

A contour of $S(Q_{ss})$ is shown in Fig. 6. In terms of $S(Q_{ss})$, the condition for nonclassicality then becomes $(1 + e)^{-1} \le S(Q_{ss}) < 1/2$, and all points to the left of the white line in Fig. 6 satisfy this condition.

The negativity of $Q_{ss}$ seen in Fig. 6 is reassuring of the previous analysis surrounding (16)—namely that the limit cycles in phase II should be regarded as nonclassical. Since the negativity of the Mandel $Q$ parameter is only a sufficient condition for nonclassicality, we would not expect it to occur for all points in phase II. What is interesting however, is that it shows a small patch of phase I to be nonclassical. Also interesting is the proof of nonclassicality for the $\wp_+ = 1$ line (excluding endpoints) using directly the definition of nonclassicality in terms of $\bar{P}_{ss}(\alpha, \alpha^*)$. These two subsets of nonclassical points, i.e. those with $\bar{P}_{ss} < 0$ and those with $Q_{ss} < 0$, when coupled with the two-photon nature of $\mathcal{L}_{\Uparrow}$, provide reasons to suspect that all points in our phase diagram might be nonclassical. However, showing this requires much more work, and take us away from our theme on pure noise-induced limit cycles in quantum theory. We thus leave the nonclassicality of the entire phase diagram here as speculation.

# 4 Nonequilibrium properties

## 4.1 Parity conservation and symmetry

The conservation of photon-number parity is a genuine nonclassical feature of our microscopic oscillator. The question is what physical consequences might this have on the phase-space flow. We find that at steady state, rotational flow in phase space (or simply circulation) is more intense for states with a greater occupancy of odd-number states. We measure the circulation intensity with orbital angular momentum in phase space, taken to be (see Appendix D)

$$\varphi \equiv \left| \Re \left[ \left\langle \hat{x} \, \mathcal{L}_{\Uparrow}^{\dagger} \, \hat{y} - \hat{y} \, \mathcal{L}_{\Uparrow}^{\dagger} \, \hat{x} \right\rangle \right] \right|, \tag{22}$$

where $\hat{x} = \hat{a} + \hat{a}^{\dagger}$, $\hat{y} = -i(\hat{a} - \hat{a}^{\dagger})$. The superoperator adjoint $\mathcal{L}_{\Uparrow}^{\dagger}$ is defined by $\text{Tr}[(\mathcal{L}_{\Uparrow}\hat{A})^{\dagger}\hat{B}] = \text{Tr}[\hat{A}^{\dagger}\mathcal{L}_{\Uparrow}^{\dagger}\hat{B}]$, for any $\hat{A}$ and $\hat{B}$. We have also defined $\Re[z] = (z + z^{*})/2$. A classical analog of (22) had been used to measure the circulation in classical stochastic limit cycles [134, 135]. We have generalised it to open quantum systems. For the noise-induced oscillator it simplifies to, for an arbitrary state (see Appendix D)

$$\varphi = \omega_0 \left\langle \hat{x}^2 + \hat{y}^2 \right\rangle. \tag{23}$$

Equation (23) is in fact still valid even if we had only a simple harmonic oscillator. The reason is because the dissipator contributions to $\varphi$ vanish. This can be understood by noting that the two-photon dissipators are rotationally symmetric. They act solely along the radial direction in phase space. Hence if $\omega_0 = 0$ (or if one transforms into a frame moving with the circular motion), then we should not see any rotational flow in phase space. This is indeed conveyed by (23). As for the appearance of $\langle \hat{x}^2 + \hat{y}^2 \rangle$ in (23), we can trace it back to the meaning of $\varphi$. We have defined $\varphi$ to measure the angular momentum in quantum phase space, but with its vectorial attribute removed. This means that the farther the orbital motion is away from the phase-space origin the larger $\varphi$ should be. Quantum mechanically, the distance away from the origin shows up as $\langle \hat{x}^2 + \hat{y}^2 \rangle$.

We can now use (23) to obtain the steady-state circulation of $\mathcal{L}_{\Uparrow}$ (see Appendix D). The result is

$$\varphi_{\text{ss}} = 4\,\omega_0 \left( \frac{2\kappa_{\Uparrow}}{\kappa_{\Downarrow} - \kappa_{\Uparrow}} + \wp_- + \frac{1}{2} \right). \tag{24}$$

The first term in $\varphi_{\text{ss}}$ describes a classical effect, while the second and third terms are quantum. The first term can be understood if the rotational flow had been purely macroscopic. Increasing the amount of noise tends to increase the average phase-space orbit (i.e. average distance away from the origin), while increasing the amount of dissipation should decrease the orbit. The competition between these two processes is captured by the first term of (24). Using the classical analog of (22), we find the macroscopic system of (9) and (10) to have a steady-state circulation given by (see Appendix D)

$$\varphi_{\text{ss}} = 4\,\omega_0 \left( \frac{2\kappa}{\Delta} \right). \tag{25}$$

This matches the first term in (24) with $\kappa$ playing the role of $\kappa_{\Uparrow}$, and $\Delta$ the role of $\kappa_{\Downarrow} - \kappa_{\Uparrow}$. This result also shows that the quantum circulation is further enhanced by a parity-sensitive term, and constant term. The constant term of $1/2$ is none other than the familiar vacuum fluctuations when the oscillator is in its lowest-energy state. As long as $\omega_0 \neq 0$, vacuum fluctuations prevent $\varphi_{\text{ss}}$ from ever reaching zero in the quantum oscillator, which is allowed classically. We also find that $\varphi_{\text{ss}} \longrightarrow \infty$ if $\kappa \longrightarrow 1$. This makes sense since on average the oscillator's amplitude increases linearly without bound when $\kappa \longrightarrow 1$ [58].

Table 1: Comparison of various physical attributes between the three oscillator models defined by $\mathcal{L}_\uparrow$ (conventional Stuart–Landau model), $\mathcal{L}_\Uparrow$ (noise-induced Stuart–Landau model), and $\mathscr{L}$ the classical analog of $\mathcal{L}_\Uparrow$ corresponding to (9). Note that Wigner negativity and probability flux are for the steady state, and parity refers to photon-number parity. Attributes that do not apply are denoted by $\varnothing$.

| Physical property | $\mathcal{L}_\uparrow$ | $\mathcal{L}_\Uparrow$ | $\mathscr{L}$ |
|---|---|---|---|
| Steady state | unique | multiple | unique |
| Limit cycle | yes | yes | no |
| Wigner negativity | no | yes | $\varnothing$ |
| Parity conservation | no | yes | $\varnothing$ |
| Parity symmetry | weak | strong | $\varnothing$ |
| Detailed balance | no | yes | yes |
| Probability flux | dissipative | conservative | conservative |

Given the existence of a conserved quantity, it is natural to ponder if there is also an underlying symmetry. Since number-parity is a discrete variable, and since our system is open, there is no unifying principle such as Noether's theorem to guide us. Nevertheless, symmetries and conservation laws have been studied for Markovian open systems [123, 136, 137], where definitions of symmetry analogous to the Hamiltonian case have been proposed [137]. A given $\mathcal{L}$ is said to possess strong symmetry, if there exists a unitary operator which commutes with its Hamiltonian and Lindblad operators. It then follows that for number parity, strong symmetry is both necessary and sufficient for its conservation, defined formally by $\mathcal{L}^\dagger (-1)^{\hat{n}} = 0$ [123]. Thus, the preceding discussion on the consequences of parity conservation on the rotational flow in $\mathcal{L}_\Uparrow$ can also be understood as consequences of parity symmetry. For comparison, we see that $\mathcal{L}_\uparrow$ does not conserve number parity, but it nevertheless has weak number-parity symmetry [123, 137]. Details and further discussions of symmetry properties are deferred to the Appendix E. The main points of our discussion are now inducted into Table 1.

## 4.2 Detailed balance and probability flux

Our noise-induced quantum limit cycle may be said to be conservative in that it is derived from a conservative system in the usual sense of nonlinear dynamics [138]. The consequences of this on the steady-state circulation were derived from (23). Here we go further and show that our noise-induced quantum limit cycle is driven by a reversible probability flux in phase space. This also entails a discussion of the closely related notion of detailed balance. We shall be referring to our results within the context of a limit cycle, but they in fact apply for a general steady state.

It is well known from classical statistical physics that a nonequilibrium steady state in detailed balance is also a steady state driven by a reversible or dissipationless probability current [139–141]. Detailed balance states that a stationary system moving from $(x_1, y_1)$ to $(x_2, y_2)$ in phase space over a time interval $\tau$, is equally likely to experience the time-reversed motion from $(\mathsf{T}[x_2], \mathsf{T}[y_2])$ to $(\mathsf{T}[x_1], \mathsf{T}[y_1])$, where the time reversal of some quantity $s$ is denoted by $\mathsf{T}[s]$. Stated formally, detailed balance is defined by

$$P_{\text{ss}}(x_2, y_2, t + \tau \, ; \, x_1, y_1, t) = P_{\text{ss}}(\mathsf{T}[x_1], \mathsf{T}[y_1], t + \tau \, ; \, \mathsf{T}[x_2], \mathsf{T}[y_2], t). \qquad (26)$$

If the nonequilibrium steady state corresponds to a limit cycle, then detailed balance says that such a limit cycle must be driven by a conservative probability current [142]. It can be shown

that our macroscopic oscillator possesses detailed balance, and indeed, we find its probability flux at steady state to be purely conservative, given by (see Appendix F)

$$\lim_{t\to\infty} \boldsymbol{J}(x,y,t) = \left[\begin{array}{c} \omega_0\, y \\ -\omega_0\, x \end{array}\right] P_{\text{ss}}(x,y). \tag{27}$$

Note the flux $\boldsymbol{J}(x,y,t)$ is defined by the continuity equation $\mathscr{L}P(x,y,t) = -\nabla\cdot\boldsymbol{J}(x,y,t)$, and $P_{\text{ss}}(x,y)$ is as in (16). The question now is whether the microscopic oscillator also has detailed balance (and hence a conservative probability flux, or vice versa). Unfortunately, there is no direct connection between detailed balance in an open quantum system and its probability flux. This makes the same problem in the quantum case more nontrivial than in the classical case. It turns out that both properties hold in the microscopic oscillator as well as shown in Table 1. They are discussed below, with the relevant proofs left to Appendix G.

In general, a Markovian open quantum system has detailed balance if and only if for any $\hat{A}$ and $\hat{B}$ [143, 144],

$$\big\langle \hat{A}(t+\tau)\hat{B}(t) \big\rangle_{\text{ss}} = \big\langle \mathsf{T}\big[\hat{B}(t+\tau)\big]\,\mathsf{T}\big[\hat{A}(t)\big] \big\rangle_{\text{ss}}, \tag{28}$$

where $\mathsf{T}[\hat{A}(t)]$ is the time-reversed $\hat{A}(t)$ [145]. One can then show that $\mathcal{L}_{\Uparrow}$, as defined by (1), indeed satisfies (28) (see Appendix G). It is again worthwhile to contrast $\mathcal{L}_{\Uparrow}$ with the conventional model of $\mathcal{L}_{\uparrow}$ in (5), which does not satisfy detailed balance, and may thus be understood to generate a dissipative limit cycle.

To show that the microscopic limit cycle has a conservative probability flux, we refer to its Wigner equation of motion in phase space, which we write as $\partial W(x,y,t)/\partial t = \mathscr{L}_{\Uparrow}W(x,y,t)$. From this we may define a Wigner current $\boldsymbol{J}_{\Uparrow}(x,y,t)$ by the continuity equation $\mathscr{L}_{\Uparrow}W(x,y,t) = -\nabla\cdot\boldsymbol{J}_{\Uparrow}(x,y,t)$ [146, 147]. The Wigner current can then be shown to satisfy (see Appendix G)

$$\lim_{t\to\infty} \boldsymbol{J}_{\Uparrow}(x,y,t) = \left[\begin{array}{c} \omega_0\, y \\ -\omega_0\, x \end{array}\right] W_{\text{ss}}(x,y), \tag{29}$$

where $W_{\text{ss}}(x,y)$ is as defined in (12)–(14). With this, we may unambiguously refer to the noise-induced quantum limit cycle simply as conservative.

## 5 Summary

Limit cycles are emblematic of regular motion in nonlinear nonequilibrium systems. In this paper we found that multiplicative quantum noise alone can induce a microscopic (i.e. quantum) damped oscillator to undergo limit-cycle oscillations. Our results are based on a simple model whose steady-state Wigner function may be derived and for which a microscopic interpretation of the multiplicative noise is possible. Such nonlinear open quantum systems are rare. This has allowed us to completely classify the noise-induced transitions, which are summarized in the phase diagrams of Figs. 4 and 5.

Our central result is the discovery of noise-induced transitions (going from $K = 0$ to $K > 0$ for a given $\wp_+$ in the phase diagram) which are both pure and genuinely nonclassical. The possibility of such noise-induced transitions in an open quantum system is consistent with the physical interpretation of $\mathcal{L}_{\Uparrow}$ in Fig. 3. We also find such noise-induced limit cycles to have fundamentally different traits from the conventional model of $\mathcal{L}_{\uparrow}$ (Table 1). Interestingly, when transiting the phase diagram from region I to II along a fixed $K$, we also find a Hopf bifurcation with respect to $\wp_+$, which is a nonclassical parameter. When $\wp_+ < 1/2$ we then enter phase III where we refer to the noise-induced limit cycle as "quantum protected," owing to their robustness against noise (as opposed to limit cycles in phase II), and the preservation of Wigner negativity at all values of $K$.

Finally, we mention again that although our microscopic (i.e. quantum) oscillator does not possess a macroscopic (i.e. classical) limit, it is still interesting to ask if a classical oscillator that best emulates the quantum oscillator might possess a pure noise-induced limit cycle. We have shown that even such a classical oscillator cannot be induced by multiplicative noise to undergo limit-cycle oscillations. This comparison between the quantum and classical models provides an alternative perspective on the nonclassicality of our quantum pure noise-induced transition, one that is independent of nonclassicality measures or witnesses.

## Acknowledgments

The authors would like to thank Jingu Pang for the illustration of the phase diagram. AC thanks Paweł Kurzyński and Ranjith Nair for their comments on the manuscript and useful discussions.

**Funding information** AC, WKM, and LCK are supported by the Ministry of Education, Singapore, and the National Research Foundation, Singapore. AC and LCK acknowledge the Ministry of Education, Singapore and the National Research Foundation, Singapore and the NRF QEP grant NRF2021-QEP2-02-P03. CN is supported by the National Research Foundation of Korea (NRF) grant funded by the Korea government (MSIT) (NRF-2022R1F1A1063053).

## A  Steady-state Wigner function

Here we wish to derive the Wigner quasiprobability distribution corresponding to the steady-state density operator

$$\rho_{ss} = \wp_+ \rho_+ + \wp_- \rho_- , \tag{A.1}$$

where

$$\wp_+ = \sum_{n=0}^{\infty} \langle 2n|\rho(0)|2n\rangle , \quad \wp_- = \sum_{n=0}^{\infty} \langle 2n+1|\rho(0)|2n+1\rangle , \tag{A.2}$$

$$\rho_+ = (1-K)\sum_{n=0}^{\infty} K^n |2n\rangle\langle 2n| , \quad \rho_- = (1-K)\sum_{n=0}^{\infty} K^n |2n+1\rangle\langle 2n+1| , \quad K = \frac{\kappa_\Uparrow}{\kappa_\Downarrow} . \tag{A.3}$$

Recall that $\rho_{ss}$ is defined by $\mathcal{L}_\Uparrow \rho_{ss} = 0$ where

$$\mathcal{L}_\Uparrow = -i\,\omega_0 [\hat{a}^\dagger\hat{a}, \cdot] + \kappa_\Downarrow \mathcal{D}[\hat{a}^2] + \kappa_\Uparrow \mathcal{D}[\hat{a}^{\dagger 2}] . \tag{A.4}$$

### A.1  Derivation in terms of complex variables $(\alpha, \alpha^*)$

The Wigner function corresponding to an arbitrary $\rho$ is defined by the following integral over the entire complex plane $\mathbb{C}$ [148],

$$\bar{W}(\alpha, \alpha^*) = \frac{1}{\pi^2} \int_{\mathbb{C}} d^2\beta\, e^{\beta^*\alpha - \beta\alpha^*} \text{Tr}\left[\rho\, e^{\beta\hat{a}^\dagger - \beta^*\hat{a}}\right] . \tag{A.5}$$

Using (A.1) in (A.5) we get,

$$\bar{W}_{ss}(\alpha, \alpha^*) = \wp_+ \bar{W}_+(\alpha, \alpha^*) + \wp_- \bar{W}_-(\alpha, \alpha^*) , \tag{A.6}$$

where $W_+$ and $W_-$ are Wigner functions corresponding to the states $\rho_+$ and $\rho_-$ respectively. Again, $W_+$ and $W_-$ are each a linear combination of Wigner functions of Fock states. It is well known that $\rho = |n\rangle\langle n|$ has the Wigner function

$$\bar{W}_n(\alpha,\alpha^*) = (-1)^n \frac{2}{\pi} e^{-2|\alpha|^2} L_n(4|\alpha|^2), \tag{A.7}$$

where $L_n(v)$ is a Laguerre polynomial in $v$ for each $n$. We thus have, from (A.5) and (A.7),

$$\bar{W}_+(\alpha,\alpha^*) = \sum_{n=0}^{\infty} K^n \bar{W}_{2n}(\alpha,\alpha^*) = \frac{2}{\pi}(1-K)e^{-2|\alpha|^2}\sum_{n=0}^{\infty} K^n L_{2n}(4|\alpha|^2), \tag{A.8}$$

$$\bar{W}_-(\alpha,\alpha^*) = \sum_{n=0}^{\infty} K^n \bar{W}_{2n+1}(\alpha,\alpha^*) = -\frac{2}{\pi}(1-K)e^{-2|\alpha|^2}\sum_{n=0}^{\infty} K^n L_{2n+1}(4|\alpha|^2). \tag{A.9}$$

The sums in (A.8) and (A.9) may be derived in closed form by using the generating function for Laguerre polynomials, given by

$$G(u,v) = \sum_{n=0}^{\infty} u^n L_n(v) = \frac{1}{1-u} e^{-uv/(1-u)}. \tag{A.10}$$

This allows us to establish

$$G(u,v) + G(-u,v) = \sum_{n=0}^{\infty} u^n L_n(v) + \sum_{n=0}^{\infty}(-1)^n u^n L_n(v) = 2\sum_{n=0}^{\infty} u^{2n} L_{2n}(v), \tag{A.11}$$

$$G(u,v) - G(-u,v) = \sum_{n=0}^{\infty} u^n L_n(v) - \sum_{n=0}^{\infty}(-1)^n u^n L_n(v) = 2\sum_{n=0}^{\infty} u^{2n+1} L_{2n+1}(v). \tag{A.12}$$

Rearranging and using (A.10) gives,

$$\sum_{n=0}^{\infty} u^{2n} L_n(v) = \frac{1}{2}\left[\frac{1}{1-u} e^{-uv/(1-u)} + \frac{1}{1+u} e^{uv/(1+u)}\right], \tag{A.13}$$

$$\sum_{n=0}^{\infty} u^{2n+1} L_{2n+1}(v) = \frac{1}{2}\left[\frac{1}{1-u} e^{-uv/(1-u)} - \frac{1}{1+u} e^{uv/(1+u)}\right]. \tag{A.14}$$

These relations can now be used to obtain $W_+$ and $W_-$ on letting

$$u = \sqrt{K}, \quad v = 4|\alpha|^2. \tag{A.15}$$

We thus arrive at the steady-state Wigner function

$$\bar{W}_{ss}(\alpha,\alpha^*) = \wp_+ \bar{W}_+(\alpha,\alpha^*) + \wp_- \bar{W}_-(\alpha,\alpha^*), \tag{A.16}$$

where

$$\bar{W}_+(\alpha,\alpha^*) = \frac{1-K}{\pi} e^{-2|\alpha|^2}\left\{\frac{1}{1-\sqrt{K}} \exp\left[-\frac{4\sqrt{K}|\alpha|^2}{1-\sqrt{K}}\right] + \frac{1}{1+\sqrt{K}} \exp\left[\frac{4\sqrt{K}|\alpha|^2}{1+\sqrt{K}}\right]\right\}, \tag{A.17}$$

$$\bar{W}_-(\alpha,\alpha^*) = \frac{1-K}{\pi\sqrt{K}} e^{-2|\alpha|^2}\left\{\frac{1}{1+\sqrt{K}} \exp\left[\frac{4\sqrt{K}|\alpha|^2}{1+\sqrt{K}}\right] - \frac{1}{1-\sqrt{K}} \exp\left[-\frac{4\sqrt{K}|\alpha|^2}{1-\sqrt{K}}\right]\right\}. \tag{A.18}$$

We can independently verify (A.16)–(A.18) by showing that it is indeed the steady-state solution of corresponding equation of motion for the Wigner function. Such an equation of

motion may be derived by noting that (A.5) implies us the following operator correspondences [148]:

$$\hat{a}\rho \longleftrightarrow \left(\alpha + \frac{1}{2}\frac{\partial}{\partial\alpha^*}\right)\bar{W}(\alpha,\alpha^*),$$ (A.19)

$$\hat{a}^\dagger\rho \longleftrightarrow \left(\alpha^* - \frac{1}{2}\frac{\partial}{\partial\alpha}\right)\bar{W}(\alpha,\alpha^*),$$ (A.20)

$$\rho\,\hat{a} \longleftrightarrow \left(\alpha - \frac{1}{2}\frac{\partial}{\partial\alpha^*}\right)\bar{W}(\alpha,\alpha^*),$$ (A.21)

$$\rho\,\hat{a}^\dagger \longleftrightarrow \left(\alpha^* + \frac{1}{2}\frac{\partial}{\partial\alpha}\right)\bar{W}(\alpha,\alpha^*).$$ (A.22)

The corresponding equation of motion for the Wigner function can then be shown to be

$$\frac{\partial}{\partial t}\bar{W}(\alpha,\alpha^*,t) \equiv \mathscr{L}_\Uparrow \bar{W}(\alpha,\alpha^*,t)$$

$$= i\omega_0\left(\frac{\partial}{\partial\alpha}\alpha - \frac{\partial}{\partial\alpha^*}\alpha^*\right)\bar{W}(\alpha,\alpha^*,t)$$

$$+ \kappa_\Downarrow\left[\frac{\partial}{\partial\alpha}\left(|\alpha|^2 - 1\right)\alpha + \frac{\partial^2}{\partial\alpha\partial\alpha^*}\left(|\alpha|^2 - \frac{1}{2}\right) + \frac{1}{4}\frac{\partial^3}{\partial\alpha^2\partial\alpha^*}\alpha\right]\bar{W}(\alpha,\alpha^*,t)$$

$$+ \kappa_\Downarrow\left[\frac{\partial}{\partial\alpha^*}\left(|\alpha|^2 - 1\right)\alpha^* + \frac{\partial^2}{\partial\alpha^*\partial\alpha}\left(|\alpha|^2 - \frac{1}{2}\right) + \frac{1}{4}\frac{\partial^3}{\partial\alpha^{*2}\partial\alpha}\alpha^*\right]\bar{W}(\alpha,\alpha^*,t)$$

$$+ \kappa_\Uparrow\left[-\frac{\partial}{\partial\alpha}\left(|\alpha|^2 + 1\right)\alpha + \frac{\partial^2}{\partial\alpha\partial\alpha^*}\left(|\alpha|^2 + \frac{1}{2}\right) - \frac{1}{4}\frac{\partial^3}{\partial\alpha^2\partial\alpha^*}\alpha\right]\bar{W}(\alpha,\alpha^*,t)$$

$$+ \kappa_\Uparrow\left[-\frac{\partial}{\partial\alpha^*}\left(|\alpha|^2 + 1\right)\alpha^* + \frac{\partial^2}{\partial\alpha^*\partial\alpha}\left(|\alpha|^2 + \frac{1}{2}\right) - \frac{1}{4}\frac{\partial^3}{\partial\alpha^{*2}\partial\alpha}\alpha^*\right]\bar{W}(\alpha,\alpha^*,t).$$ (A.23)

We then find explicitly on substituting $\bar{W}_{ss}(\alpha,\alpha^*)$ into (A.23) that

$$\mathscr{L}_\Uparrow \bar{W}_{ss}(\alpha,\alpha^*) = 0.$$ (A.24)

## A.2 Polar coordinates $(r,\phi)$

It will be convenient to reparameterise $\bar{W}_{ss}$ in terms of polar coordinates for ease of comparison to the classical steady-state distribution later on. The complex variable $\alpha$ is then related to polar coordinates $(r,\phi)$ by

$$\alpha = r\exp(i\phi).$$ (A.25)

It is then simple to show that

$$\int_{\mathbb{C}} d^2\alpha\,\bar{W}_{ss}(\alpha,\alpha^*) = \int_0^\infty dr\int_0^{2\pi} d\phi\, r\,\bar{W}_{ss}(re^{i\phi},re^{-i\phi}) = 1.$$ (A.26)

Thus the new Wigner function is

$$\tilde{W}_{ss}(r,\phi) = r\,\bar{W}_{ss}(re^{i\phi},re^{-i\phi}) = \wp_+\tilde{W}_+(r,\phi) + \wp_-\tilde{W}_-(r,\phi),$$ (A.27)

where

$$\tilde{W}_+(r,\phi) = \frac{1-K}{\pi}\,r\,e^{-2r^2}\left\{\frac{1}{1-\sqrt{K}}\exp\left[-\frac{4\sqrt{K}r^2}{1-\sqrt{K}}\right] + \frac{1}{1+\sqrt{K}}\exp\left[\frac{4\sqrt{K}r^2}{1+\sqrt{K}}\right]\right\},$$ (A.28)

$$\tilde{W}_-(r,\phi) = \frac{1-K}{\pi\sqrt{K}}\,r\,e^{-2r^2}\left\{\frac{1}{1+\sqrt{K}}\exp\left[\frac{4\sqrt{K}r^2}{1+\sqrt{K}}\right] - \frac{1}{1-\sqrt{K}}\exp\left[-\frac{4\sqrt{K}r^2}{1-\sqrt{K}}\right]\right\}.$$ (A.29)

### A.3 Cartesian coordinates $(x, y)$

The Cartesian coordinates are often the most intuitive for visualizing the dynamics and steady states. Here we define the Cartesian coordinates $(x, y)$ by

$$x = 2r\cos\phi, \quad y = 2r\sin\phi. \tag{A.30}$$

It is then simple to show

$$\int_0^\infty dr \int_0^{2\pi} d\phi\, \tilde{W}_{ss}(r, \phi) = \int_{-\infty}^\infty dx \int_{-\infty}^\infty dy\, \frac{1}{2\sqrt{x^2 + y^2}} \tilde{W}_{ss}(r(x, y), \phi(x, y)) = 1. \tag{A.31}$$

where

$$r = \frac{1}{2}\sqrt{x^2 + y^2}, \quad \phi = \arctan\left(\frac{y}{x}\right). \tag{A.32}$$

As before,

$$W_{ss}(x, y) = \frac{1}{2\sqrt{x^2 + y^2}} \tilde{W}_{ss}(r(x, y), \phi(x, y)) = \wp_+ W_+(x, y) + \wp_- W_-(x, y), \tag{A.33}$$

with

$$W_+(x, y) = \frac{1 - K}{4\pi} e^{-(x^2 + y^2)/2} \left\{ \frac{1}{1 - \sqrt{K}} \exp\left[ -\frac{\sqrt{K}(x^2 + y^2)}{1 - \sqrt{K}} \right] + \frac{1}{1 + \sqrt{K}} \exp\left[ \frac{\sqrt{K}(x^2 + y^2)}{1 + \sqrt{K}} \right] \right\}, \tag{A.34}$$

$$W_-(x, y) = \frac{1 - K}{4\pi\sqrt{K}} e^{-(x^2 + y^2)/2} \left\{ \frac{1}{1 + \sqrt{K}} \exp\left[ \frac{\sqrt{K}(x^2 + y^2)}{1 + \sqrt{K}} \right] - \frac{1}{1 - \sqrt{K}} \exp\left[ -\frac{\sqrt{K}(x^2 + y^2)}{1 - \sqrt{K}} \right] \right\}. \tag{A.35}$$

## B  Classification of noise-induced transitions

Here we derive the different noise-induced transitions when thermal noise is added to the oscillator. Each type of transition is defined by the steady-state behavior of the Wigner function in the presence of noise. We show how the $(\wp_+, K)$ plane can be divided into three different regions, each corresponding to a distinct phase of the Wigner function.

### B.1  Phase diagram

As can be seen from the steady-state Wigner function in any of the three coordinates above, it is a function of only the radial distance from the origin. There is no loss of generality in treating the Wigner distribution as a single-variable function. We thus define the single-variable function $W(r)$ from either (A.16)–(A.18) or (A.33)–(A.35) to be the unnormalized Wigner function,

$$W(r) \equiv \bar{W}_{ss}(re^{i\phi}, re^{-i\phi}) = 4W_{ss}(2r\cos\phi, 2r\sin\phi). \tag{B.1}$$

Below we work with $W(r)$, for which single-variable calculus applies. The function $W(r)$ can exhibit different qualitative behaviors depending on the parameters $K$ and $\wp_+$. Using P-bifurcations, the existence of a quantum limit cycle is defined by the value of

$$r_\star \equiv \arg\max W(r). \tag{B.2}$$

If $r_\star = 0$, the unnormalized Wigner function has a single peak only at the origin, reflecting the stable fixed point at the origin. If on the other hand $r_\star > 0$, the unnormalized Wigner function has a degenerate maxima along a circle of radius $r_\star$ in phase space, reflecting stable limit-cycle behaviour. To compute the transition point, we first solve $W'(r) = 0$ for $r$, where the prime denotes differentiation with respect to the argument. We then find a trivial solution $r = 0$, and a nontrivial solution

$$r_\star^2 = \frac{1-K}{8\sqrt{K}} \ln \left\{ \frac{(1+\sqrt{K})^4 [1-(1-K)\wp_+ - \sqrt{K}]}{(1-\sqrt{K})^4 [1-(1-K)\wp_+ + \sqrt{K}]} \right\}, \tag{B.3}$$

which corresponds to the limit cycle radius. Imposing the condition $r_\star^2 > 0$ for the limit-cycle solution, we find the condition for a limit cycle to exist is

$$\wp_+ < \frac{3+K}{4(1+K)}. \tag{B.4}$$

This equation now defines the border between regions I and II in Figs. 4 and 5 in the main text. The same result can also be obtained by demanding $W''(0) > 0$. Note the limit-cycle transition occurs for all critical points $(K^c, \wp_+^c)$ satisfying $4\wp_+^c = (3+K^c)/(1+K^c)$. The leading order behavior of $r_\star$ near $(K^c, \wp_+^c)$ can be calculated as

$$r_\star \approx \frac{4}{1-K^c} \sqrt{K^c - K} + \frac{2\sqrt{2}}{2\wp_+^c - 1} \sqrt{\wp_+^c - \wp_+}. \tag{B.5}$$

The square-root scaling law for the limit cycle amplitude $r_\star$ is characteristic of a supercritical Hopf bifurcation [149].

The Wigner function $\bar{W}_{ss}(\alpha, \alpha^*)$ in (A.16) cannot be negative without it being negative at the origin. To see this, we use the rotational symmetry of $W_{ss}(x, y)$ and consider $W_{ss}(x, 0)$ for $x \geq 0$ without loss of generality. Suppose now $W_{ss}(x, 0)$ contains negative values for some $x_* > 0$, we then have

$$\frac{\wp_+ - (1-\wp_+)/\sqrt{K}}{1-\sqrt{K}} \exp\left(-\frac{\sqrt{K}}{1-\sqrt{K}} x_*^2\right) + \frac{\wp_+ + (1-\wp_+)/\sqrt{K}}{1+\sqrt{K}} \exp\left(\frac{\sqrt{K}}{1+\sqrt{K}} x_*^2\right) < 0. \tag{B.6}$$

Upon rearranging gives

$$\exp\left(\frac{2\sqrt{K}}{1-K} x_*^2\right) < \frac{(1-\wp_+)/\sqrt{K} - \wp_+}{1-\sqrt{K}} \left[ \frac{1+\sqrt{K}}{\wp_+ + (1-\wp_+)/\sqrt{K}} \right], \tag{B.7}$$

for some $x_* > 0$. Since the left-hand side is monotonically increasing in $x$, this condition must also be satisfied for all $0 \leq x \leq x_*$. Hence $W(r)$ contains negative values if and only if $W(0)$ is negative. From this we obtain the condition for Wigner negativity to be

$$\wp_+ < \frac{1}{2}. \tag{B.8}$$

This equation now defines the border between regions II and III in Figs. 4 and 5 in the main text. Summarizing, we obtain three qualitatively distinct phases of solutions in the $(K, \wp_+)$ parameter space:

- Phase I: No limit cycle and no Wigner negativity.

- Phase II: Limit cycle with a positive Wigner function.

- Phase III: Limit cycle with a negative Wigner function.

## B.2 Example: Coherent initial state

To illustrate the ideas developed in the previous section, let us consider a specific example of an initial coherent state $|\alpha\rangle$. The even and odd steady-state populations are

$$\wp_+ = e^{-|\alpha|^2} \cosh|\alpha|^2, \quad \wp_- = e^{-|\alpha|^2} \sinh|\alpha|^2. \tag{B.9}$$

Applying the limit cycle condition to coherent states, we obtain

$$|\alpha|^2 > \frac{1}{2} \ln\left[\frac{2(1+K)}{1-K}\right]. \tag{B.10}$$

From this we learn that if we add an infinite amount of external noise to the system (i.e. $K \longrightarrow 1$), then the oscillator must also possess an infinite amount of energy (i.e. $|\alpha|^2$) if a limit cycle is to be induced.

Recall that for the Wigner function to exhibit negativity, we must have $\wp_+ < 1/2$. From Eq. (B.9), it can be easily seen that $\wp_+ > 1/2$. In other words, it is impossible to induce a negative Wigner function by initializing in any coherent state. This can actually be extended to any state with a positive Wigner function (including Gaussian states) as follows: If $\bar{W}(\alpha, \alpha^*) > 0$, then $\bar{W}(0,0) \propto (\wp_+ - \wp_-) > 0$, which implies $\wp_+ > 1/2$. Since number parity is conserved, the Wigner function at the origin remains positive in the steady state. Moreover, the steady state Wigner function in (A.16) precludes any negativity without $\bar{W}(0,0) < 0$, hence the entire Wigner function remains positive in the steady state. Note the converse is not true, i.e. a $\rho(0)$ with a negative Wigner function may have $\wp_+ > 1/2$. An example is the even cat state.

## B.3 Tail behavior

At large distances from the origin, the steady-state Wigner function is asymptotic to the un-normalized Gaussian

$$W_G(x,y) = \frac{1-\sqrt{K}}{4\pi\sqrt{K}} \left[1-(1-\sqrt{K})\wp_+\right] \exp\left[-\frac{1-\sqrt{K}}{2(1+\sqrt{K})}(x^2+y^2)\right]. \tag{B.11}$$

Denoting the total area under $W_G(x,y)$ as $A$, we find that

$$\frac{1+\sqrt{K}}{2} \le A = \frac{1+\sqrt{K}}{2\sqrt{K}} \left[1-(1-\sqrt{K})\wp_+\right] \le \frac{1+\sqrt{K}}{2\sqrt{K}}. \tag{B.12}$$

This shows that $A \longrightarrow 1$ as $K \longrightarrow 1$. This implies that the state becomes more Gaussian-like in the high-excitation limit, which is physically intuitive.

## C Classical steady-state probability density

Here we solve for the steady-state probability density function for the classical system defined by the Itô stochastic differential equation,

$$d\alpha(t) = \left[-i\omega_0\alpha(t) + 2\kappa\alpha(t) - \Delta|\alpha(t)|^2\alpha(t)\right]dt + \alpha^*(t)dW(t), \tag{C.1}$$

where $\omega_0$ is the frequency of the free oscillations and $\Delta > 0$. As in the main text, $dW(t)$ is a complex Wiener increment satisfying

$$dW^*(t)dW(t) = 4\kappa\,dt. \tag{C.2}$$

## C.1 Polar coordinates $(R, \Phi)$

A major simplification occurs if we convert from the complex-variable description to polar coordinates.

$$\alpha(t) = R(t) e^{i\Phi(t)}. \tag{C.3}$$

With the exception of $\alpha$, we denote random processes using capital letters and their realizations using the corresponding small letter. The stochastic dynamics of $R(t)$ and $\Phi(t)$ may then be derived using standard techniques [150]. They are given by

$$dR(t) = \left[ 3\kappa R(t) - \Delta R^3(t) \right] dt + \frac{R(t)}{2} dW_R(t), \tag{C.4}$$

$$d\Phi(t) = -\omega_0 dt + \frac{1}{2} dW_\Phi(t), \tag{C.5}$$

where $dW_R(t)$ and $dW_\Phi(t)$ are independent real Wiener increments obeying the following Itô rules

$$dW_R(t) dW_\Phi(t) = 0, \tag{C.6}$$

$$\left[ dW_R(t) \right]^2 = \left[ dW_\Phi(t) \right]^2 = 8\kappa dt. \tag{C.7}$$

From (C.4) and (C.5) we can see that the dynamics of $R(t)$ and $\Phi(t)$ are independent processes and our two-dimensional system simplifies to two one-dimensional systems. This independence of $R(t)$ and $\Phi(t)$ means that each process has its own Fokker–Planck equation. For $R(t)$, it is given by

$$\frac{\partial}{\partial t} P_R(r,t) \equiv \mathscr{L}_R P_R(r,t) = -\frac{\partial}{\partial r} \left( 3\kappa r - \Delta r^3 \right) P_R(r,t) + \frac{1}{2} \frac{\partial^2}{\partial r^2} 2\kappa r^2 P_R(r,t). \tag{C.8}$$

This permits a closed-form solution for the radial steady-state distribution,

$$\wp_R(r) \equiv \lim_{t\to\infty} P_R(r,t) = \frac{\Delta}{\kappa} r\, e^{-\Delta r^2/2\kappa}. \tag{C.9}$$

Note this has the form of a Rayleigh distribution. Similarly the phase dynamics in (C.5) corresponds to the operator $\mathscr{L}_\Phi$

$$\frac{\partial}{\partial t} P_\Phi(\phi,t) \equiv \mathscr{L}_\Phi P_\Phi(\phi,t) = -\frac{\partial}{\partial \phi} (-\omega_0) P_\Phi(\phi,t) + \frac{1}{2} \frac{\partial^2}{\partial \phi^2} 2\kappa P_\Phi(\phi,t). \tag{C.10}$$

Imposing periodic boundary conditions (suitable for a circular variable such as $\Phi$) on a $2\pi$ interval gives

$$\wp_\Phi(\phi) \equiv \lim_{t\to\infty} P_\Phi(\phi,t) = \frac{1}{2\pi}. \tag{C.11}$$

Since the radial and phase motions are independent, we have $\tilde{P}(r,\phi,t) = P_R(r,t)P_\Phi(\phi,t)$ whose evolution can be obtained by adding the operators $\mathscr{L}_R$ and $\mathscr{L}_\Phi$,

$$\frac{\partial}{\partial t} \tilde{P}(r,\phi,t) \equiv \tilde{\mathscr{L}} \tilde{P}(r,\phi,t) = (\mathscr{L}_R + \mathscr{L}_\Phi) \tilde{P}(r,\phi,t). \tag{C.12}$$

The joint steady-state distribution for $R(t)$ and $\Phi(t)$ is therefore simply

$$\tilde{P}_{\mathrm{ss}}(r,\phi) \equiv \lim_{t\to\infty} \tilde{P}(r,\phi,t) \tag{C.13}$$

$$= \wp_R(r)\wp_\Phi(\phi) = \frac{\Delta}{2\pi\kappa} r\, e^{-\Delta r^2/2\kappa}. \tag{C.14}$$

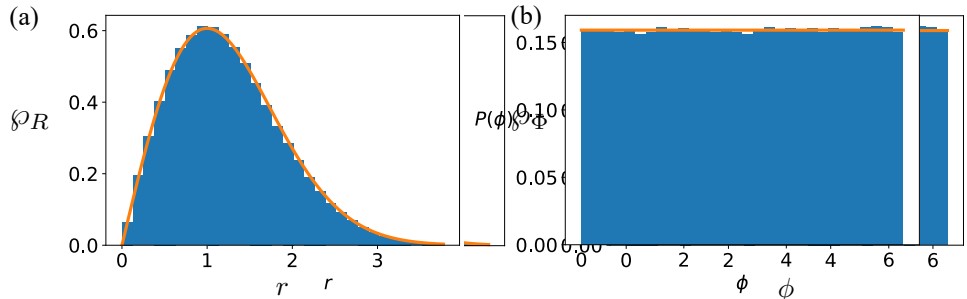

Figure 7: Histograms from numerical simulations of (C.4) and (C.5) for $\kappa = \Delta = 1$, $\omega_0 = 10$ with $10^6$ samples. Since we are only interested in the steady state, the transient dynamics in the stochastic simulations are discarded. (a) Radial probability density (ideally a Rayleigh distribution, shown by the orange line). (b) Phase probability density (ideally uniform on a $2\pi$ interval, orange line).

We have also numerically verified (C.14) by simulating the Itô stochastic differential equations (C.4) and (C.5). An example of the sampled distributions are shown in Fig. 7 (see figure caption for parameter values). Note from this result we can already see a qualitative difference between the classical and quantum systems. The classical steady-state distribution lacks the exponential growth present in $\tilde{W}_{ss}(r, \phi)$. A well-known property of the Rayleigh distribution is that it is the probability density for the modulus of a complex random variable whose real and imaginary parts are independent and identically distributed Gaussians with zero mean. Thus, the form of (C.9) already tells us that $\alpha(t)$ is described by two independent processes in phase space.

## C.2 Cartesian coordinates $(X, Y)$

We can directly convert (C.14) to Cartesian coordinates. We define here $(X, Y)$ as earlier

$$X(t) = 2R(t)\cos[\Phi(t)], \quad Y(t) = 2R(t)\sin[\Phi(t)]. \tag{C.15}$$

As with the Wigner function,

$$R(t) = \frac{1}{2}\sqrt{X^2(t) + Y^2(t)}, \quad \Phi(t) = \arctan\left[\frac{Y(t)}{X(t)}\right], \tag{C.16}$$

and we obtain at once,

$$P_{ss}(x, y) = \frac{1}{2\sqrt{x^2 + y^2}} \tilde{P}_{ss}(r(x, y), \phi(x, y)) = \frac{\Delta}{8\pi\kappa} e^{-\Delta(x^2 + y^2)/8\kappa}. \tag{C.17}$$

We can verify that this is indeed the correct probability distribution in phase space by directly substituting this back into the Fokker–Planck equation for $X(t)$ and $Y(t)$. We define here

$$\alpha(t) = \frac{1}{2}[X(t) + iY(t)], \tag{C.18}$$

so that on taking the real and imaginary parts of (C.1) we get

$$dX(t) = \left\{\omega_0 Y(t) + 2\kappa X(t) - \frac{\Delta}{4}\left[X^2(t) + Y^2(t)\right]X(t)\right\}dt + \frac{1}{2}\left[X(t)\,dW_X(t) + Y(t)\,dW_Y(t)\right], \tag{C.19}$$

$$dY(t) = \left\{-\omega_0 X(t) + 2\kappa Y(t) - \frac{\Delta}{4}\left[X^2(t) + Y^2(t)\right]Y(t)\right\}dt + \frac{1}{2}\left[X(t)\,dW_Y(t) - Y(t)\,dW_X(t)\right]. \tag{C.20}$$

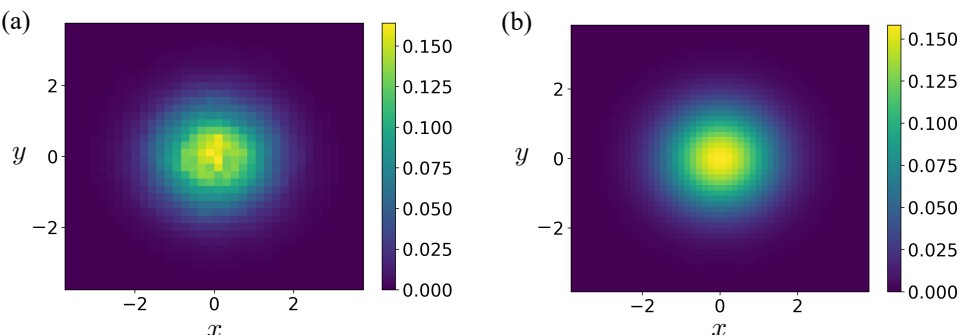

Figure 8: (a) Density plot for the Cartesian probability distribution $P_{ss}(x, y)$ generated from (C.19) and (C.20) for $\kappa = \Delta = 1, \omega_0 = 10$ with $10^6$ samples. The transient dynamics is discarded. (b) Density plot for the analytical Gaussian distribution in (C.17).

The noise terms in (C.19) and (C.20) arise from decomposing $dW(t)$ into its real and imaginary parts in a similar fashion as $\alpha(t)$,

$$dW(t) = \frac{1}{2} \left[ dW_X(t) + i \, dW_Y(t) \right], \tag{C.21}$$

where $dW_X(t)$ and $dW_Y(t)$ are independent real Wiener increments

$$\left[ dW_X(t) \right]^2 = \left[ dW_Y(t) \right]^2 = 8 \kappa \, dt. \tag{C.22}$$

The Fokker–Planck equation corresponding to (C.19) and (C.20) is

$$\frac{\partial}{\partial t} P(x, y, t) \equiv \mathscr{L} P(x, y, t) \tag{C.23}$$

$$= -\frac{\partial}{\partial x} \left[ \omega_0 y + 2\kappa x - \frac{\Delta}{4} (x^2 + y^2) x \right] P(x, y, t) + \frac{1}{2} \frac{\partial^2}{\partial x^2} 2\kappa (x^2 + y^2) P(x, y, t)$$

$$- \frac{\partial}{\partial y} \left[ -\omega_0 x + 2\kappa y - \frac{\Delta}{4} (x^2 + y^2) y \right] P(x, y, t) + \frac{1}{2} \frac{\partial^2}{\partial y^2} 2\kappa (x^2 + y^2) P(x, y, t). \tag{C.24}$$

We then find from (C.17) and (C.23) that

$$\mathscr{L} P_{ss}(x, y) = 0. \tag{C.25}$$

This is also numerically verified in Fig. 8 using (C.19) and (C.20) from which we see the sampled $P_{ss}(x, y)$ shows good agreement with the exact Gaussian distribution. It is worth pointing out here that we have also checked the consistency between the stochastic differential equations in polar coordinates against those in Cartesian coordinates by plotting $(X, Y)$ as $(2R \cos \Phi, 2R \sin \Phi)$. The steady-state distribution for the latter is again a Gaussian as expected. It is generally useful to simulate stochastic differential equations as they provide some intuition for the processes of interest via direct visualization. Although we have not shown such results here, a good way to proceed is to use (C.4) and (C.5) instead of (C.19) and (C.20) as the former pair of equations are decoupled.

# D Rotational flow in quantum phase space

## D.1 Definition

The goal here is to generalise the measure of circulation from Refs. [134, 152] to an open quantum system. To motivate the generalisation to quantum mechanics we begin with a deterministic classical system defined by

$$\frac{d}{dt} x = f(x, y), \quad \frac{d}{dt} y = g(x, y). \tag{D.1}$$

If the phase-space point has circular motion then we can expect that it should have a nonvanishing angular momentum in phase space. It thus makes sense to define an angular momentum in phase space in analogous fashion to the orbital angular momentum of a mechanical point particle, except now the position and velocity vectors are given by their phase-space analogues. Using an orthonormal basis $\{e_x, e_y\}$ in Cartesian coordinates, we may then define the phase-space position vector $\boldsymbol{u} = x\,\boldsymbol{e}_x + y\,\boldsymbol{e}_y$, and phase-space velocity vector $\boldsymbol{v} = f(x, y)\boldsymbol{e}_x + g(x, y)\boldsymbol{e}_y$. We then define the angular-momentum vector as the cross product,

$$\boldsymbol{u} \times \boldsymbol{v} = [x\,g(x, y) - y\,f(x, y)](\boldsymbol{e}_x \times \boldsymbol{e}_y). \tag{D.2}$$

In fact, we will not be interested in $\boldsymbol{u} \times \boldsymbol{v}$ as a vector quantity, so we will simply define

$$\varphi \equiv \left\| \boldsymbol{u} \times \boldsymbol{v} \right\| = \left| x\,g(x, y) - y\,f(x, y) \right|. \tag{D.3}$$

If the system is noisy, so that $x(t)$ and $y(t)$ become random processes $X(t)$ and $Y(t)$, then an average over the realisations of $X(t)$ and $Y(t)$ may be performed as a sensible generalisation of (D.3),

$$\varphi \equiv \left| \mathrm{E}\!\left[ X\,g(X, Y) - Y\,f(X, Y) \right] \right|, \tag{D.4}$$

where $\mathrm{E}[f(X, Y)]$ denotes a classical ensemble average of $f(X, Y)$ against $P(x, y, t)$. Note that if we add multiplicative white noise to (D.1), then (D.4) assumes that $g(X, Y)$ and $f(X, Y)$ correspond to the Stratonovich forms of (D.1), either by directly interpreting (D.1) as Stratonovich equations or by finding the equivalent Stratonovich forms.

A further generalisation of $\varphi$ to quantum mechanics is then possible on letting $X \longrightarrow \hat{x}$ and $Y \longrightarrow \hat{y}$, except that upon quantization, $\hat{x}$ and $\hat{y}$ become canonically conjugate, satisfying

$$[\hat{x}, \hat{y}] = 2i\,\hat{\mathbb{1}}. \tag{D.5}$$

However, quantization also entails that we choose a particular ordering between $\hat{x}$ and $\hat{y}$ in such a way that $\hat{x}$ and $\hat{y}$ remain Hermitian under time evolution. This results in the new functions $\check{f}(\hat{x}, \hat{y})$ and $\check{g}(\hat{x}, \hat{y})$ respectively. By the same token, we define $\varphi$ in quantum mechanics by the following symmetrized form

$$\varphi \equiv \frac{1}{2} \left| \left\langle \left[ \hat{x}\,\check{g}(\hat{x}, \hat{y}) + \check{g}(\hat{x}, \hat{y})\hat{x} \right] - \left[ \hat{y}\,\check{f}(\hat{x}, \hat{y}) + \check{f}(\hat{x}, \hat{y})\hat{y} \right] \right\rangle \right|. \tag{D.6}$$

This ensures that $\varphi$ is real valued, as it should be. If the system has a generator of time evolution given by $\mathcal{L}$, i.e. $d\rho(t)/dt = \mathcal{L}\rho(t)$, then we replace $\check{f}(\hat{x}, \hat{y})$ and $\check{g}(\hat{x}, \hat{y})$ by using the adjoint of $\mathcal{L}$, defined with respect to the Hilbert–Schmidt inner product,

$$\mathrm{Tr}\!\left[ \hat{A}^{\dagger} \mathcal{L}^{\dagger} \hat{B} \right] = \mathrm{Tr}\!\left[ (\mathcal{L}\hat{A})^{\dagger} \hat{B} \right], \tag{D.7}$$

we therefore arrive at

$$\varphi \equiv \left| \Re\!\left[ \left\langle \hat{x}\,\mathcal{L}^{\dagger}\hat{y} - \hat{y}\,\mathcal{L}^{\dagger}\hat{x} \right\rangle \right] \right|, \tag{D.8}$$

where $\Re[z] = (z + z^*)/2$.

## D.2 General formula for the microscopic oscillator

Here we wish to derive $\varphi$ for the noise-induced oscillator defined by

$$\mathcal{L}_\Uparrow = -i\,\omega_0\,[\hat{a}^\dagger\hat{a},\cdot] + \kappa_\Downarrow\mathcal{D}[\hat{a}^2] + \kappa_\Uparrow\mathcal{D}[\hat{a}^{\dagger 2}]. \tag{D.9}$$

It is straightforward to show that

$$\mathcal{L}_\Uparrow^\dagger = i\,\omega_0\,[\hat{a}^\dagger\hat{a},\cdot] + \kappa_\Downarrow\left(\mathcal{D}[\hat{a}^2]\right)^\dagger + \kappa_\Uparrow\left(\mathcal{D}[\hat{a}^{\dagger 2}]\right)^\dagger, \tag{D.10}$$

where

$$\left(\mathcal{D}[\hat{a}^2]\right)^\dagger = \hat{a}^{\dagger 2}\cdot\hat{a}^2 - \frac{1}{2}\,\hat{a}^{\dagger 2}\hat{a}^2\cdot - \frac{1}{2}\cdot\hat{a}^{\dagger 2}\hat{a}^2, \tag{D.11}$$

$$\left(\mathcal{D}[\hat{a}^{\dagger 2}]\right)^\dagger = \hat{a}^2\cdot\hat{a}^{\dagger 2} - \frac{1}{2}\,\hat{a}^2\hat{a}^{\dagger 2}\cdot - \frac{1}{2}\cdot\hat{a}^2\hat{a}^{\dagger 2}. \tag{D.12}$$

The expectation value in (D.8) becomes

$$\left\langle\hat{x}\,\mathcal{L}^\dagger\hat{y} - \hat{y}\,\mathcal{L}^\dagger\hat{x}\right\rangle = i\,\omega_0\left\langle\hat{x}\,[\hat{a}^\dagger\hat{a},\hat{y}] - \hat{y}\,[\hat{a}^\dagger\hat{a},\hat{x}]\right\rangle + \kappa_\Uparrow\left\langle\hat{x}\left(\mathcal{D}[\hat{a}^{\dagger 2}]\right)^\dagger\hat{y} - \hat{y}\left(\mathcal{D}[\hat{a}^{\dagger 2}]\right)^\dagger\hat{x}\right\rangle$$
$$+ \kappa_\Downarrow\left\langle\hat{x}\left(\mathcal{D}[\hat{a}^2]\right)^\dagger\hat{y} - \hat{y}\left(\mathcal{D}[\hat{a}^2]\right)^\dagger\hat{x}\right\rangle. \tag{D.13}$$

As we explained in the main text, an intuitive understanding of the dissipators in phase space suggests that they do not contribute to $\varphi$. This can be shown by writing $\hat{x}$ and $\hat{y}$ in terms of $\hat{a}$ and $\hat{a}^\dagger$. For the terms proportional to $\kappa_\Uparrow$ we have,

$$\left\langle\hat{x}\left(\mathcal{D}[\hat{a}^{\dagger 2}]\right)^\dagger\hat{y} - \hat{y}\left(\mathcal{D}[\hat{a}^{\dagger 2}]\right)^\dagger\hat{x}\right\rangle = -i\left\langle(\hat{a}+\hat{a}^\dagger)\left(\mathcal{D}[\hat{a}^{\dagger 2}]\right)^\dagger(\hat{a}-\hat{a}^\dagger)\right\rangle \tag{D.14}$$
$$+ i\left\langle(\hat{a}-\hat{a}^\dagger)\left(\mathcal{D}[\hat{a}^{\dagger 2}]\right)^\dagger(\hat{a}+\hat{a}^\dagger)\right\rangle$$
$$= i\,2\left\langle\hat{a}\left(\mathcal{D}[\hat{a}^{\dagger 2}]\right)^\dagger\hat{a}^\dagger\right\rangle - i\,2\left\langle\hat{a}^\dagger\left(\mathcal{D}[\hat{a}^{\dagger 2}]\right)^\dagger\hat{a}\right\rangle. \tag{D.15}$$

Similarly, the terms proportional to $\kappa_\Downarrow$ follow on replacing $\hat{a}^{\dagger 2}$ by $\hat{a}^2$ in the dissipator,

$$\left\langle\hat{x}\left(\mathcal{D}[\hat{a}^2]\right)^\dagger\hat{y} - \hat{y}\left(\mathcal{D}[\hat{a}^2]\right)^\dagger\hat{x}\right\rangle = i\,2\left\langle\hat{a}\left(\mathcal{D}[\hat{a}^2]\right)^\dagger\hat{a}^\dagger\right\rangle - i\,2\left\langle\hat{a}^\dagger\left(\mathcal{D}[\hat{a}^2]\right)^\dagger\hat{a}\right\rangle. \tag{D.16}$$

The expectation values in (D.15) and (D.16) now contain equal numbers of $\hat{a}$ and $\hat{a}^\dagger$ which means that ultimately they can be written in terms of the Hermitian operator $\hat{n} = \hat{a}^\dagger\hat{a}$. They are thus purely imaginary and vanish on substitution into the definition of $\varphi$ in (D.8). For the sake of concreteness we state their exact forms here,

$$\left\langle\hat{x}\left(\mathcal{D}[\hat{a}^{\dagger 2}]\right)^\dagger\hat{y} - \hat{y}\left(\mathcal{D}[\hat{a}^{\dagger 2}]\right)^\dagger\hat{x}\right\rangle = i\,4(\langle\hat{n}\rangle + 1), \tag{D.17}$$

$$\left\langle\hat{x}\left(\mathcal{D}[\hat{a}^2]\right)^\dagger\hat{y} - \hat{y}\left(\mathcal{D}[\hat{a}^2]\right)^\dagger\hat{x}\right\rangle = -i\,4\langle\hat{n}\rangle. \tag{D.18}$$

The expression for $\varphi$ therefore simplifies to

$$\varphi = \omega_0\left\langle\hat{x}^2 + \hat{y}^2\right\rangle. \tag{D.19}$$

## D.3 Steady-state formula for the microscopic oscillator

We can now derive an explicit formula for the steady-state circulation by using our result for $\rho_{ss}$. Since the steady state is diagonal in the number basis, it is more convenient to reexpress (D.19) as

$$\varphi = 4\,\omega_0\left(\langle\hat{n}\rangle + \frac{1}{2}\right). \tag{D.20}$$

Note the 1/2 in the parentheses represents a vacuum contribution to the phase-space circulation. The steady-state average photon number is then, upon using (A.1),

$$\langle \hat{n} \rangle_{\text{ss}} = 2\left(1-K\right)\sum_{n=0}^{\infty} n\,K^n + 2\,\wp_-\left(1-K\right)\sum_{n=0}^{\infty} K^n, \tag{D.21}$$

where we have used $\wp_+ + \wp_- = 1$. The second sum is simply a geometric series while it is simple to show that the first sum is given by

$$\sum_{n=0}^{\infty} n\,K^n = \frac{K}{(1-K)^2}. \tag{D.22}$$

Equation (D.21) therefore becomes

$$\langle \hat{n} \rangle_{\text{ss}} = \frac{2K}{1-K} + \wp_-. \tag{D.23}$$

Substituting this back into (D.20) we thus arrive at an expression for the steady-state circulation $\varphi_{\text{ss}}$

$$\varphi_{\text{ss}} = 4\,\omega_0\left(\frac{2K}{1-K} + \wp_- + \frac{1}{2}\right). \tag{D.24}$$

We may also express $\varphi_{\text{ss}}$ as a function of only either $\wp_0$ or $\wp_1$, where $\wp_n = \langle n|\rho_{\text{ss}}|n\rangle$. Choosing here to write it as a function of $\wp_0$ we note that $\wp_-$ may be written as

$$\wp_- = \frac{1-K-\wp_0}{1-K}, \tag{D.25}$$

where we have used (or see Refs. [90, 154]),

$$\wp_+ = \frac{\wp_0}{1-K}, \quad \wp_- = \frac{\wp_1}{1-K}. \tag{D.26}$$

Substituting (D.25) into (D.24) then gives

$$\varphi_{\text{ss}} = 4\,\omega_0\left[\frac{1+K-\wp_0}{1-K} + \frac{1}{2}\right]. \tag{D.27}$$

### D.4 Steady-state formula for the macroscopic oscillator

The definition of $\varphi$ for a classical system was already discussed en route to the quantum-mechanical definition in (D.4). Using the stochastic differential equations in (C.19) and (C.20) it is trivial to see that the time-dependent circulation is

$$\varphi = \omega_0\,\text{E}\left[X^2(t) + Y^2(t)\right]. \tag{D.28}$$

The steady-state value then follows simply by noting that $X^2(t) + Y^2(t) = 4R^2(t)$, and that the statistical moments for the Rayleigh distribution are well documented. For a Rayleigh distribution in the form of (C.9), the steady-state mean and variance are

$$\text{E}_{\text{ss}}\left[R(t)\right] = \sqrt{\frac{\pi\kappa}{2\Delta}}, \quad \text{V}_{\text{ss}}\left[R(t)\right] = \text{E}_{\text{ss}}\left[R^2(t)\right] - \left\{\text{E}_{\text{ss}}\left[R(t)\right]\right\}^2 = \frac{(4-\pi)\kappa}{2\Delta}. \tag{D.29}$$

We thus have

$$\varphi_{\text{ss}} = 4\,\omega_0\left(\text{V}_{\text{ss}}\left[R(t)\right] - \left\{\text{E}_{\text{ss}}\left[R(t)\right]\right\}^2\right) = 4\,\omega_0\left(\frac{2\kappa}{\Delta}\right). \tag{D.30}$$

# E    Parity symmetry in the microscopic oscillator

Arguably no discussion of a conserved quantity can be considered complete without at least mentioning its associated symmetry. Thus we devote this section to some details and some further discussions related to the symmetry properties of our microscopic model in (D.9). For a symmetry operation represented by some unitary operator $\hat{U}$, we can distingush between two types of symmetries [155]. We begin our discussion by recalling what they are from the literature. The first is called a strong symmetry. This requires that for a general Lindbladian

$$\mathcal{L} = -i\,[\hat{H}, \cdot\,] + \sum_{k=1}^{M} \gamma_k\, \mathcal{D}[\hat{c}_k], \quad \gamma_k \geq 0, \forall\, k, \tag{E.1}$$

$\hat{U}$ satisfies

$$[\,\hat{U}, \hat{H}\,] = [\,\hat{U}, \hat{c}_k\,] = 0, \quad \forall\, k. \tag{E.2}$$

This can be understood to generalize the symmetry condition for Hamiltonian systems [defined by (E.1) with $\gamma_k = 0$ for all values of $k$] to the case when dissipative processes are present. Of course, $\mathcal{L}$ is a generator of time evolution for a Markovian quantum system just as $\hat{H}$ is for a closed system. It thus also makes sense to define symmetry for an open system by requiring that the action of $\hat{U}$ commute with the Lindbladian, i.e.

$$[\mathcal{U}, \mathcal{L}] = 0, \tag{E.3}$$

where $\mathcal{U}$ is defined by $\mathcal{U}\rho = \hat{U}\rho\,\hat{U}^{-1}$. If we find a $\hat{U}$ that satisfies (E.3), it is said to be a weak symmetry. Strong symmetry implies weak symmetry but not vice versa [137, 156].

Using the above, we can show that $\mathcal{L}_\Uparrow$ possesses a strong symmetry corresponding to photon-number parity, defined by $\hat{\Pi} = (-1)^{\hat{n}}$ where $\hat{n} = \hat{a}^\dagger\hat{a}$. It is simple to see that $\hat{\Pi} = \hat{\Pi}^\dagger = \hat{\Pi}^{-1}$. Since $\hat{\Pi}$ is a function of the number operator it commutes with the Hamiltonian in (D.9). The only nontrivial requirements are from (E.2) with $\hat{c}_1 = \hat{a}^2$ and $\hat{c}_2 = \hat{a}^{\dagger 2}$,

$$[\hat{\Pi}, \hat{a}^2] = [\hat{\Pi}, \hat{a}^{\dagger 2}] = 0. \tag{E.4}$$

This is simple to show and has been discussed in the context of two-photon absorption (i.e. $\kappa_\Uparrow = 0$) [156]. As noted in the main text, parity conservation and strong symmetry are equivalent [156]. For a general $\mathcal{L}$, conservation of an arbitrary quantity represented by $\hat{C}$ is defined formally as

$$\mathcal{L}^\dagger\hat{C} = 0. \tag{E.5}$$

We may thus use the condition of strong symmetry to show that $\mathcal{L}_\uparrow$ does not conserve photon-number parity. We recall for convenience here that $\mathcal{L}_\uparrow$ is given by

$$\mathcal{L}_\uparrow = -i\,\omega_0\,[\hat{a}^\dagger\hat{a}, \cdot\,] + \kappa_\Downarrow\, \mathcal{D}[\hat{a}^2] + \kappa_\uparrow\, \mathcal{D}[\hat{a}^\dagger]. \tag{E.6}$$

This is already intuitive from the appearance of $\mathcal{D}[\hat{a}^\dagger]$ in $\mathcal{L}_\uparrow$, since one-photon transitions take odd-parity number states to even-parity ones and vice versa. The corresponding mathematical statement is simply $[\hat{\Pi}, \hat{a}^\dagger] = -2\,\hat{a}^\dagger\hat{\Pi} \neq 0$. The possibility of $\mathcal{L}_\uparrow$ to satisfying weak number-parity symmetry remains open. However, instead of showing this directly, here we point out that both $\mathcal{L}_\uparrow$ and $\mathcal{L}_\Uparrow$ satisfy continuous rotational symmetry for which parity symmetry is a special case of. A continuous rotation has the unitary operator $\hat{P}_\phi = \exp(-i\phi\,\hat{n})$ where $\phi$ is a

continuous real-valued parameter. It is not difficult to show that the operation of a rotation in phase space commutes with either $\mathcal{L}_\uparrow$ and $\mathcal{L}_\Uparrow$,

$$[\mathcal{P}_\phi, \mathcal{L}_\uparrow] = [\mathcal{P}_\phi, \mathcal{L}_\Uparrow] = 0, \tag{E.7}$$

where $\mathcal{P}_\phi \rho = \hat{P}_\phi \rho \hat{P}_\phi^{-1}$. This property was in fact shown in Ref. [157] for $\mathcal{L}_\Uparrow$ but was referred to as phase covariance. Clearly photon-number parity transformation corresponds to a rotation with $\phi = \pi$. Thus, both $\mathcal{L}_\uparrow$ and $\mathcal{L}_\Uparrow$ exhibit a weak continuous symmetry defined by $\hat{P}_\phi$, and consequently a weak discrete symmetry given by $\hat{P}_\pi = \hat{\Pi}$ (noting of course that $\mathcal{L}_\Uparrow$ actually exhibits a strong symmetry as well).

Note that we have expressed $\rho_{\rm ss}$ in (A.1) deliberately as a linear combination of a state with even parity $\rho_+$, and a state with odd parity $\rho_-$. This is a very natural decomposition of $\rho_{\rm ss}$ given that photon-number parity is conserved. Its form makes the steady state simple to see if the initial state does not contain either even or odd number states. The normalization of $\rho_{\rm ss}$ is also trivial when expressed in the form of (A.1). There is a closely related idea, in fact a theorem, which decomposes $\rho_{\rm ss}$ not in terms of states like (A.1), but in terms of an orthonormal operator basis. The expansion coefficients in this decomposition are defined by averages of conserved quantities with respect to the initial state [156]. We complete our discussion of the symmetry and conservation of parity by simply finding this an expansion for $\rho_{\rm ss}$. Given an initial state $\rho(0)$, and an $\mathcal{L}$ with no purely imaginary eigenvalues, Ref. [156] has shown that the steady state may be expanded in terms of $D$ linearly independent conserved quantities $\{\hat{C}_k\}_{k=0}^{D-1}$ in the following form

$$\rho_{\rm ss} = \sum_{k=0}^{D-1} {\rm Tr}\big[\hat{C}_k^\dagger \rho(0)\big]\, \hat{M}_k, \tag{E.8}$$

where $\{\hat{M}_k\}_{k=0}^{D-1}$ is an orthonormal basis with respect to the Hilbert–Schmidt inner product, i.e. ${\rm Tr}[\hat{M}_j^\dagger \hat{M}_k] = \delta_{j,k}$.

To show that $\rho_{\rm ss}$ for $\mathcal{L}_\Uparrow$ can be put in the form of (E.8), we note that it has two linearly independent conserved quantities, namely the parity of even and odd photon numbers. Hence $D = 2$. The expansion in (E.8) may then be achieved with

$$\hat{C}_0 = \frac{\sqrt{1-K}}{2}\big(\hat{1} + \hat{\Pi}\big), \quad \hat{C}_1 = \frac{\sqrt{1-K}}{2}\big(\hat{1} - \hat{\Pi}\big). \tag{E.9}$$

These operators are orthogonal since they contain only nonoverlapping projectors in the Fock basis. Orthogonality then implies linear independence. To see that they are conserved we note that $\mathcal{L}_\Uparrow^\dagger \hat{\Pi} = \mathcal{L}_\Uparrow^\dagger \hat{1} = 0$. These can be shown straightforwardly from (D.10)–(D.11). It then follows from the linearity of $\mathcal{L}_\Uparrow^\dagger$ that $\mathcal{L}_\Uparrow^\dagger \hat{C}_0 = \mathcal{L}_\Uparrow^\dagger \hat{C}_1 = 0$. The associated operator basis is then

$$\hat{M}_0 = \sqrt{1-K} \sum_{n=0}^\infty K^n\, |2n\rangle\langle 2n|, \quad \hat{M}_1 = \sqrt{1-K} \sum_{n=0}^\infty K^n\, |2n+1\rangle\langle 2n+1|. \tag{E.10}$$

Clearly $\hat{M}_0$ and $\hat{M}_1$ are clearly orthogonal to each other,

$$\mathrm{Tr}\big[\hat{M}_0^\dagger \hat{M}_1\big] = \mathrm{Tr}\big[\hat{M}_1^\dagger \hat{M}_0\big] = 0. \tag{E.11}$$

It is also straightforward to see that they are normalized,

$$\mathrm{Tr}\big[\hat{M}_0^\dagger \hat{M}_0\big] = \mathrm{Tr}\big[\hat{M}_1^\dagger \hat{M}_1\big] = 1. \tag{E.12}$$

One may also verify that the steady state written in the form of (E.8) using (E.9) and (E.10) is indeed normalized. Note that $\hat{M}_0$ and $\hat{M}_1$ are positive and Hermitian operators but do not have unit trace. We mention also that (E.8) applies in the case of $\kappa_{\Uparrow} = \kappa = 0$ as well. However, in this case there is an additional conserved quantity arising from the coherences as discussed in the main text, so that $D = 3$. As we will not be using this, the reader is referred to Ref. [156] for the exact expression for the conserved quantity.

## F Classical detailed balance

### F.1 Definition

Let us now use the classical model to build some intuition about detailed balance and the nature of the probability current. A classical stochastic system with two degrees of freedom is said to possess detailed balance if the following relation is satisfied at steady state,

$$P_{\text{ss}}(x_2, y_2, t + \tau; x_1, y_1, t) = P_{\text{ss}}(\mathsf{T}[x_1], \mathsf{T}[y_1], t + \tau; \mathsf{T}[x_2], \mathsf{T}[y_2], t). \tag{F.1}$$

Here we are defining $\mathsf{T}[x] = \pi_X x$ and $\mathsf{T}[y] = \pi_Y y$ (also valid on replacing $x$ by $X$ and $y$ by $Y$), and $\pi_X$, $\pi_Y$ may be $\pm 1$ depending on whether $X$ and $Y$ are even ($+1$) or odd ($-1$) variables under time reversal. It can then be shown that a classical Markovian system given by $\partial P(x, y, t)/\partial t = \mathscr{L} P(x, y, t)$ satisfies detailed balance if and only if [139, 159]

$$P_{\text{ss}}(x, y) = P_{\text{ss}}(\mathsf{T}[x], \mathsf{T}[y]), \tag{F.2}$$

where $\mathsf{T}$ denotes the operation of time reversal and

$$P_{\text{ss}}(x, y) \mathscr{L}^{\dagger}(x, y) = \mathscr{L}(\mathsf{T}[x], \mathsf{T}[y]) P_{\text{ss}}(x, y). \tag{F.3}$$

We have also written out the dependence of $\mathscr{L}$ on $x$ and $y$ explicitly in order to define its time-reversed version. For real functions on $\mathbb{R}^2$, the adjoint of $\mathscr{L}$ is defined by the inner product

$$\int_{-\infty}^{\infty} dx \int_{-\infty}^{\infty} dy \, f(x, y) \mathscr{L}^{\dagger} g(x, y) = \int_{-\infty}^{\infty} dx \int_{-\infty}^{\infty} dy \, g(x, y) \mathscr{L} f(x, y). \tag{F.4}$$

For our macroscopic oscillator, $\mathscr{L}$ is defined by a Fokker–Planck equation in terms of drift vector $\boldsymbol{A}$, and a diffusion matrix $D$,

$$\boldsymbol{A}(x, y) = \begin{bmatrix} A_X(x, y) \\ A_Y(x, y) \end{bmatrix}, \quad D(x, y) = \begin{bmatrix} D_X(x, y) & D_{XY}(x, y) \\ D_{YX}(x, y) & D_Y(x, y) \end{bmatrix}. \tag{F.5}$$

These can be read off from (C.23), but for the purpose of this section, we work with the general form of $\mathscr{L}$, given by

$$\mathscr{L} = -\frac{\partial}{\partial x} A_X(x, y) - \frac{\partial}{\partial y} A_Y(x, y) + \frac{1}{2} \left[ \frac{\partial^2}{\partial x^2} D_X(x, y) + 2 \frac{\partial^2}{\partial x \partial y} D_{XY}(x, y) + \frac{\partial^2}{\partial y^2} D_Y(x, y) \right]. \tag{F.6}$$

Note the diffusion matrix is always symmetric so that $D_{XY}(x, y) = D_{YX}(x, y)$. Proving (F.2) and (F.3) to be true using (F.6) would not add any insight to our understanding of the microscopic oscillator. For us, the significance of (F.2) and (F.3) is that they have counterparts in quantum theory, as will be seen later. In the classical theory, they are also the necessary and sufficient conditions for the the steady-state probability current to be purely reversible [139, 159]. In

fact, conditions (F.2) and (F.3) for a general Markov process are satisfied if and only if at steady state, the probability flux is reversible and divergenceless, while the diffusion matrix transforms under time reversal as

$$D_X(x,y) = \pi_X^2 \, D_X(\mathsf{T}[x], \mathsf{T}[y]), \quad D_Y(z) = \pi_Y^2 \, D_Y(\mathsf{T}[x], \mathsf{T}[y]), \tag{F.7}$$

$$D_{XY}(x,y) = \pi_X \, \pi_Y \, D_{XY}(\mathsf{T}[x], \mathsf{T}[y]). \tag{F.8}$$

The separation of the probability current into reversible and irreversible components follow from a formal decomposition of the drift into reversible and irreversible parts. These depend on the time reversal properties of the drift vector, which are defined by

$$\overleftrightarrow{A}(x,y) = \frac{1}{2}\left[A(x,y) - \Pi A(\mathsf{T}[x], \mathsf{T}[y])\right], \quad \vec{A}(x,y) = \frac{1}{2}\left[A(x,y) + \Pi A(\mathsf{T}[x], \mathsf{T}[y])\right]. \tag{F.9}$$

We have labelled the reversible drift using a bidirectional arrow and the irreversible drift by a unidirectional arrow. The matrix $\Pi$ is simply,

$$\Pi = \begin{bmatrix} \pi_X & 0 \\ 0 & \pi_Y \end{bmatrix}. \tag{F.10}$$

The probability current, which we denote by $J$, is defined by writing the Fokker–Planck equation as a continuity equation for the probability density,

$$\frac{\partial}{\partial t} P(x,y,t) = -\nabla \cdot J(x,y,t). \tag{F.11}$$

We may then decompose the probability current by using (F.9) into reversible and irreversible parts,

$$J(x,y,t) = \overleftrightarrow{J}(x,y,t) + \vec{J}(x,y,t). \tag{F.12}$$

They are simply

$$\overleftrightarrow{J}(x,y,t) = \overleftrightarrow{A}(x,y) P(x,y,t), \quad \vec{J}(x,y,t) = \vec{A}(x,y) P(x,y,t) - \frac{1}{2}\left[\nabla^\top D(x,y) P(x,y,t)\right]^\top, \tag{F.13}$$

where $S^\top$ denotes the matrix transpose of $S$. As is usual, the steady-state current may be formally defined as

$$j(x,y) = \lim_{t\to\infty} J(x,y,t). \tag{F.14}$$

The condition for the steady-state probability current to be reversible and divergenceless can then be stated as

$$\nabla \cdot \overleftrightarrow{j}(x,y) = 0, \quad \vec{j}(x,y) = \mathbf{0}. \tag{F.15}$$

We may simply use (F.13) and replace the time-dependent probability density by its steady-state value. As mentioned earlier, condition (F.15) along with (F.7) and (F.8) are equivalent to (F.2) and (F.3).

## F.2 Detailed balance in the macroscopic oscillator

The task of showing that our macroscopic oscillator satisfies (F.7), (F.8), and (F.15) is now a simple matter. Since we are thinking of the macroscopic variables $X$ and $Y$ as the classical limits of $\hat{x}$ and $\hat{y}$, these would have to be defined as even and odd variables under time reversal to be consistent with quantum mechanics [160]. Hence,

$$\pi_X = 1, \quad \pi_Y = -1. \tag{F.16}$$

The drift vector and diffusion matrix from (C.23) are

$$A(x, y) = \begin{bmatrix} \omega_0 y + 2\kappa x - \Delta(x^2 + y^2)x/4 \\ -\omega_0 x + 2\kappa y - \Delta(x^2 + y^2)y/4 \end{bmatrix}, \quad D(x, y) = \begin{bmatrix} 2\kappa(x^2 + y^2) & 0 \\ 0 & 2\kappa(x^2 + y^2) \end{bmatrix}. \tag{F.17}$$

Clearly, $D(x, y)$ satisfies (F.7) and (F.8). Recall that we have also shown in (C.17) the steady-state distribution of the macroscopic oscillator to be

$$P_{ss}(x, y) = \frac{\Delta}{8\pi\kappa} e^{-\Delta(x^2 + y^2)/8\kappa}. \tag{F.18}$$

From these one find (F.15) to be true, and in particular, with

$$\overleftrightarrow{j}(x, y) = \begin{bmatrix} \omega_0 y \\ -\omega_0 x \end{bmatrix} P_{ss}(x, y). \tag{F.19}$$

# G  Quantum detailed balance

## G.1  Definition

A Markovian quantum system defined by $d\rho(t)/dt = \mathcal{L}\rho(t)$ is said to be in detailed balance if and only if [143, 162],

$$\left\langle \hat{A}(t + \tau)\hat{B}(t) \right\rangle_{ss} = \left\langle \mathsf{T}\left[\hat{B}(t + \tau)\right] \mathsf{T}\left[\hat{A}(t)\right] \right\rangle_{ss}, \quad \forall \hat{A}, \hat{B}, \tag{G.1}$$

where $\mathsf{T}$ denotes the operation of time reversal via an antiunitary and antilinear operator $\hat{T}$ [160]. It maps operators to operators,

$$\mathsf{T}(\hat{A}) = \hat{T}\hat{A}^\dagger \hat{T}^{-1}, \quad \forall \hat{A}, \tag{G.2}$$

and scalars to their complex conjugates, i.e. $\hat{T} z \hat{T}^{-1} = z^*$ for $z \in \mathbb{C}$. This condition of quantum detailed balance was first proposed in Ref. [161], and rigorously justified in Ref. [162]. It is more general than detailed balance in the sense of a Pauli equation. The latter is a semiclassical condition and is implied by (G.1). It can then be shown that if the steady state is time-reversal invariant, i.e.

$$\rho_{ss} = \mathsf{T}(\rho_{ss}), \tag{G.3}$$

then (G.1) is implied by the following superoperator condition

$$\rho_{ss}\mathcal{L}^\dagger = \mathsf{T}(\mathcal{L})\rho_{ss}. \tag{G.4}$$

The time-reversed Lindbladian $\mathsf{T}(\mathcal{L})$ is another superoperator defined such that,

$$\mathsf{T}(\mathcal{L}\hat{A}) = \mathsf{T}(\mathcal{L})\mathsf{T}(\hat{A}), \quad \forall \hat{A}. \tag{G.5}$$

Thus if both (G.3) and (G.4) are true for a given $\mathcal{L}$ then quantum detailed balance is proven. Conditions (G.3) and (G.4) are analogs of (F.2) and (F.3), which is why we are using (G.1) to prove quantum detailed balance. But note that unlike (F.2) and (F.3) in the classical theory, (G.3) and (G.4) are only sufficient conditions for quantum detailed balance. For the microscopic oscillator given by $\mathcal{L}_{\Uparrow}$, the steady state is diagonal in the number basis, and since number states are time-reversal invariant, it follows that (G.3) holds. We thus only need to check (G.4) which requires the time-reversed Lindbladian.

### G.2 Time-reversed Lindbladian

Here we derive the time-reversed Lindbladian for the microscopic oscillator of $\mathcal{L}_{\Uparrow}$. Recall for convenience that $\mathcal{L}_{\Uparrow}$ is given by

$$\mathcal{L}_{\Uparrow} = -i\,\omega_0\,[\hat{a}^\dagger\hat{a},\cdot\,] + \kappa_{\Downarrow}\left(\hat{a}^2\cdot\hat{a}^{\dagger 2} - \frac{1}{2}\,\hat{a}^{\dagger 2}\,\hat{a}^2\cdot - \frac{1}{2}\cdot\hat{a}^{\dagger 2}\hat{a}^2\right)$$
$$+ \kappa_{\Uparrow}\left(\hat{a}^{\dagger 2}\cdot\hat{a}^2 - \frac{1}{2}\,\hat{a}^2\,\hat{a}^{\dagger 2}\cdot - \frac{1}{2}\cdot\hat{a}^2\hat{a}^{\dagger 2}\right). \tag{G.6}$$

Using (G.2), we therefore have for an arbitrary $\hat{A}$,

$$\mathsf{T}(\mathcal{L}_{\Uparrow}\hat{A}) = -i\,\omega_0\left[\mathsf{T}(\hat{A})\,\mathsf{T}(\hat{a})\,\mathsf{T}(\hat{a}^\dagger) - \mathsf{T}(\hat{a})\,\mathsf{T}(\hat{a}^\dagger)\,\mathsf{T}(\hat{A})\right]$$
$$+ \kappa_{\Downarrow}\left[\mathsf{T}(\hat{a}^{\dagger 2})\,\mathsf{T}(\hat{A})\,\mathsf{T}(\hat{a}^2) - \frac{1}{2}\,\mathsf{T}(\hat{A})\,\mathsf{T}(\hat{a}^2)\,\mathsf{T}(\hat{a}^{\dagger 2}) - \frac{1}{2}\,\mathsf{T}(\hat{a}^2)\,\mathsf{T}(\hat{a}^{\dagger 2})\,\mathsf{T}(\hat{A})\right]$$
$$+ \kappa_{\Uparrow}\left[\mathsf{T}(\hat{a}^2)\,\mathsf{T}(\hat{A})\,\mathsf{T}(\hat{a}^{\dagger 2}) - \frac{1}{2}\,\mathsf{T}(\hat{A})\,\mathsf{T}(\hat{a}^{\dagger 2})\,\mathsf{T}(\hat{a}^2) - \frac{1}{2}\,\mathsf{T}(\hat{a}^{\dagger 2})\,\mathsf{T}(\hat{a}^2)\,\mathsf{T}(\hat{A})\right]. \tag{G.7}$$

To work out $\mathsf{T}(\hat{a})$ and $\mathsf{T}(\hat{a}^\dagger)$ we can write $\hat{a} = (\hat{x} + i\hat{y})/2$ and use the fact that $[\hat{x},\hat{y}] = i\,2\,\hat{1}$ enforces $\hat{x}$ to be an even operator, and $\hat{y}$ an odd operator under time reversal [160]:

$$\mathsf{T}(\hat{x}) = \hat{x}, \quad \mathsf{T}(\hat{y}) = -\hat{y}. \tag{G.8}$$

Using (G.8), the time-reversed annihilation and creation operators are then

$$\mathsf{T}(\hat{a}) = \frac{1}{2}\left[\mathsf{T}(\hat{x}) - i\,\mathsf{T}(\hat{y})\right] = \hat{a}^\dagger, \quad \mathsf{T}(\hat{a}^\dagger) = \frac{1}{2}\left[\mathsf{T}(\hat{x}) + i\,\mathsf{T}(\hat{y})\right] = \hat{a}. \tag{G.9}$$

We may then simplify (G.7) further since

$$\mathsf{T}(\hat{a}^2) = \mathsf{T}(\hat{a})\mathsf{T}(\hat{a}) = \hat{a}^{\dagger 2}, \quad \mathsf{T}(\hat{a}^{\dagger 2}) = \mathsf{T}(\hat{a}^\dagger)\mathsf{T}(\hat{a}^\dagger) = \hat{a}^2. \tag{G.10}$$

Using (G.5) we arrive at

$$\mathsf{T}(\mathcal{L}_{\Uparrow})\,\mathsf{T}(\hat{A}) = -i\,\omega_0\left[\mathsf{T}(\hat{A}),\hat{a}^\dagger\hat{a}\right] + \kappa_{\Downarrow}\,\mathcal{D}\!\left[\hat{a}^2\right]\mathsf{T}(\hat{A}) + \kappa_{\Uparrow}\,\mathcal{D}\!\left[\hat{a}^{\dagger 2}\right]\mathsf{T}(\hat{A}). \tag{G.11}$$

Note that $\mathsf{T}(\mathcal{L}_{\Uparrow})$ is not a time-reversed Lindbladian in the sense that it captures time-reversed motion of the microscopic oscillator, which is what time-reversal means in physics. In phase space, the time-reversed motion of $\mathcal{L}_{\Uparrow}$ should interchange motion in the positive $x$ direction with motion in the negative $x$ direction, and similarly for $y$. Thus time-reversed motion should interchange amplification with dissipation, and counterclockwise rotation with clockwise rotation. Although $\mathsf{T}(\mathcal{L}_{\Uparrow})$ as defined by (G.5) does not correspond to time-reversal in this sense, it is nevertheless what is required mathematically by quantum detailed balance.

### G.3 Detailed balance in the microscopic oscillator

We are now in position to prove quantum detailed balance by using (G.4). Recall also from (D.10)–(D.12) that

$$
\begin{aligned}
\mathcal{L}^\dagger =& i\,\omega_0\,[\hat{a}^\dagger\hat{a},\cdot\,] + \kappa_\Uparrow\left(\hat{a}^2\cdot\hat{a}^{\dagger 2} - \frac{1}{2}\,\hat{a}^2\,\hat{a}^{\dagger 2}\cdot - \frac{1}{2}\cdot\hat{a}^2\,\hat{a}^{\dagger 2}\right) \\
&+ \kappa_\Downarrow\left(\hat{a}^{\dagger 2}\cdot\hat{a}^2 - \frac{1}{2}\,\hat{a}^{\dagger 2}\,\hat{a}^2\cdot - \frac{1}{2}\cdot\hat{a}^{\dagger 2}\,\hat{a}^2\right).
\end{aligned}
\tag{G.12}
$$

Equation (G.4) can then be written as, using (G.11) and (G.12),

$$
\begin{aligned}
\rho_{ss}\,\mathcal{L}^\dagger - \mathsf{T}(\mathcal{L})\rho_{ss} =& \, i\,\omega_0\,[\rho_{ss},\hat{a}^\dagger\hat{a}]\cdot + \frac{1}{2}\,\kappa_\Uparrow\,[\hat{a}^2\hat{a}^{\dagger 2},\rho_{ss}]\cdot + \frac{1}{2}\,\kappa_\Downarrow\,[\hat{a}^{\dagger 2}\hat{a}^2,\rho_{ss}]\cdot \\
&+ \left(\kappa_\Uparrow\,\rho_{ss}\,\hat{a}^2 - \kappa_\Downarrow\,\hat{a}^2\rho_{ss}\right)\cdot\hat{a}^{\dagger 2} + \left(\kappa_\Downarrow\,\rho_{ss}\,\hat{a}^{\dagger 2} - \kappa_\Uparrow\,\hat{a}^{\dagger 2}\rho_{ss}\right)\cdot\hat{a}^2.
\end{aligned}
\tag{G.13}
$$

Note that since $\rho_{ss}$ is diagonal in the number basis, it must commute with $\hat{n}=\hat{a}^\dagger\hat{a}$. Now since $\hat{a}^2\hat{a}^{\dagger 2}$ and $\hat{a}^{\dagger 2}\hat{a}^2$ may also be written in terms of $\hat{n}$, all commutator terms in (G.13) vanish,

$$
[\hat{a}^\dagger\hat{a},\rho_{ss}] = [\hat{a}^2\hat{a}^{\dagger 2},\rho_{ss}] = [\hat{a}^{\dagger 2}\hat{a}^2,\rho_{ss}] = 0.
\tag{G.14}
$$

It therefore remains to show that the second line of (G.13) vanishes. Using $\rho_{ss}=\wp_+\rho_+ +\wp_-\rho_-$ [recall (A.1)–(A.3)], we have

$$
\begin{aligned}
\kappa_\Downarrow\,\hat{a}^2\rho_{ss} =& \,\wp_+\left(1-\frac{\kappa_\Uparrow}{\kappa_\Downarrow}\right)\sum_{n=1}^\infty \frac{\kappa_\Uparrow^n}{\kappa_\Downarrow^{n-1}}\,\sqrt{(2n)(2n-1)}\,|2n-2\rangle\langle 2n| \\
&+ \wp_-\left(1-\frac{\kappa_\Uparrow}{\kappa_\Downarrow}\right)\sum_{n=1}^\infty \frac{\kappa_\Uparrow^n}{\kappa_\Downarrow^{n-1}}\,\sqrt{(2n+1)(2n)}\,|2n-1\rangle\langle 2n+1|
\end{aligned}
\tag{G.15}
$$

$$
\begin{aligned}
=& \,\wp_+\left(1-\frac{\kappa_\Uparrow}{\kappa_\Downarrow}\right)\sum_{m=0}^\infty \frac{\kappa_\Uparrow^{m+1}}{\kappa_\Downarrow^{m}}\,\sqrt{(2m+2)(2m+1)}\,|2m\rangle\langle 2m+2| \\
&+ \wp_-\left(1-\frac{\kappa_\Uparrow}{\kappa_\Downarrow}\right)\sum_{m=0}^\infty \frac{\kappa_\Uparrow^{m+1}}{\kappa_\Downarrow^{m}}\,\sqrt{(2m+3)(2m+2)}\,|2m+1\rangle\langle 2m+3|
\end{aligned}
\tag{G.16}
$$

$$
= \kappa_\Uparrow\,\rho_{ss}\,\hat{a}^2,
\tag{G.17}
$$

where we have let $n=m+1$ in the second equality. Similarly,

$$
\begin{aligned}
\kappa_\Uparrow\,\hat{a}^{\dagger 2}\rho_{ss} =& \,\wp_+\left(1-\frac{\kappa_\Uparrow}{\kappa_\Downarrow}\right)\sum_{n=0}^\infty \frac{\kappa_\Uparrow^{n+1}}{\kappa_\Downarrow^{n}}\,\sqrt{(2n+1)(2n+2)}\,|2n+2\rangle\langle 2n| \\
&+ \wp_-\left(1-\frac{\kappa_\Uparrow}{\kappa_\Downarrow}\right)\sum_{n=0}^\infty \frac{\kappa_\Uparrow^{n+1}}{\kappa_\Downarrow^{n}}\,\sqrt{(2n+2)(2n+3)}\,|2n+3\rangle\langle 2n+1|
\end{aligned}
\tag{G.18}
$$

$$
\begin{aligned}
=& \,\wp_+\left(1-\frac{\kappa_\Uparrow}{\kappa_\Downarrow}\right)\sum_{m=1}^\infty \frac{\kappa_\Uparrow^{m}}{\kappa_\Downarrow^{m-1}}\,\sqrt{(2m-1)(2m)}\,|2m\rangle\langle 2m-2| \\
&+ \wp_-\left(1-\frac{\kappa_\Uparrow}{\kappa_\Downarrow}\right)\sum_{m=1}^\infty \frac{\kappa_\Uparrow^{m}}{\kappa_\Downarrow^{m-1}}\,\sqrt{(2m)(2m+1)}\,|2m+1\rangle\langle 2m-1|
\end{aligned}
\tag{G.19}
$$

$$
= \kappa_\Downarrow\,\rho_{ss}\,\hat{a}^{\dagger 2},
\tag{G.20}
$$

where this time we have let $n=m-1$ in the second equality. Hence we have shown that (G.4) holds which implies the existence of quantum detailed balance as defined by (G.1).

## G.4  Wigner current in the microscopic oscillator

We have just shown quantum detailed balance in a manner that closely matches classical detailed balance. From this, one might guess that the underlying probability flux for the microscopic oscillator to also be purely reversible as in the macroscpic case. Unfortunately we have no result that directly connects the quantum probability flux in phase space to detailed balance. Hence the proof that a purely reversible current is responsible for the microscopic oscillator at steady state has to be carried out independently. We are of course motivated by the intuition developed from the analyses above.

To find the probability flux in quantum phase space we need the equation of motion for the Wigner function. Subsequently we will refer to the probability flux as a Wigner current (but keeping in mind that it would not be a true probability current if the Wigner function becomes negative). Recall that the Wigner equation of motion was given in (A.23), but in terms of the complex coordinates. Here we convert this equation as a function of the Cartesian coordinates. This can be accomplished by noting the correspondence $\bar{W}_{ss}(\alpha, \alpha^*) \longleftrightarrow 4 W_{ss}(x, y)$ on reparameterizing the Wigner function, and also the following correspondences for differential operators

$$\frac{\partial}{\partial \alpha} \longleftrightarrow \frac{\partial}{\partial x} - i \frac{\partial}{\partial y}, \quad \frac{\partial}{\partial \alpha^*} \longleftrightarrow \frac{\partial}{\partial x} + i \frac{\partial}{\partial y}, \tag{G.21}$$

$$\frac{\partial^2}{\partial \alpha \partial \alpha^*} = \frac{\partial^2}{\partial \alpha^* \partial \alpha} \longleftrightarrow \frac{\partial^2}{\partial x^2} + \frac{\partial^2}{\partial y^2}, \tag{G.22}$$

$$\frac{\partial^3}{\partial \alpha^2 \partial \alpha^*} \longleftrightarrow \frac{\partial^3}{\partial x^3} + \frac{\partial^3}{\partial x \partial y^2} - i \frac{\partial^3}{\partial y \partial x^2} - i \frac{\partial^3}{\partial y^3}, \tag{G.23}$$

$$\frac{\partial^3}{\partial \alpha^{*2} \partial \alpha} \longleftrightarrow \frac{\partial^3}{\partial x^3} + \frac{\partial^3}{\partial x \partial y^2} + i \frac{\partial^3}{\partial y \partial x^2} + i \frac{\partial^3}{\partial y^3}, \tag{G.24}$$

where we have noted $\alpha = (x + iy)/2$. The Wigner equation of motion in terms of Cartesian coordinates is then given by

$$\begin{aligned}
\mathscr{L}_{\Uparrow} = {} & \frac{\partial}{\partial x}\left[ -\omega_0 y - (\kappa_{\Uparrow} + \kappa_{\Downarrow})x + \frac{1}{4}(\kappa_{\Downarrow} - \kappa_{\Uparrow})(x^2 + y^2)x \right] \\
& + \frac{\partial}{\partial y}\left[ \omega_0 x - (\kappa_{\Uparrow} + \kappa_{\Downarrow})y + \frac{1}{4}(\kappa_{\Downarrow} - \kappa_{\Uparrow})(x^2 + y^2)y \right] \\
& + \frac{\partial^2}{\partial x^2}\left[ \frac{1}{2}(\kappa_{\Downarrow} + \kappa_{\Uparrow})(x^2 + y^2) - (\kappa_{\Downarrow} - \kappa_{\Uparrow}) \right] \\
& + \frac{\partial^2}{\partial y^2}\left[ \frac{1}{2}(\kappa_{\Downarrow} + \kappa_{\Uparrow})(x^2 + y^2) - (\kappa_{\Downarrow} - \kappa_{\Uparrow}) \right] + \frac{\partial^3}{\partial x \partial y^2}\left[ \frac{1}{4}(\kappa_{\Downarrow} - \kappa_{\Uparrow})x \right] \\
& + \frac{\partial^3}{\partial x^3}\left[ \frac{1}{4}(\kappa_{\Downarrow} - \kappa_{\Uparrow})x \right] + \frac{\partial^3}{\partial y \partial x^2}\left[ \frac{1}{4}(\kappa_{\Downarrow} - \kappa_{\Uparrow})y \right] \\
& + \frac{\partial^3}{\partial y^3}\left[ \frac{1}{4}(\kappa_{\Downarrow} - \kappa_{\Uparrow})y \right].
\end{aligned} \tag{G.25}$$

This allows us to write the Wigner equation of motion in the form of a continuity equation. The associated current shall be denoted by $\boldsymbol{J}_{\Uparrow}$, and referred to as the Wigner current, defined by [146, 164]

$$\mathscr{L}_{\Uparrow} W(x, y, t) = -\nabla \cdot \boldsymbol{J}_{\Uparrow}(x, y, t). \tag{G.26}$$

Note that (G.25) does not give us a Fokker–Planck equation for $W(x, y, t)$ due to the presence of third-order derivatives. Therefore we have no simple procedure for decomposing the

probability current as in the classical theory of detailed balance. However, we can still use the classical theory as a guide. We thus define

$$\boldsymbol{J}_{\Uparrow}(x,y,t) = \overleftrightarrow{\boldsymbol{J}}_{\Uparrow}(x,y,t) + \vec{\boldsymbol{J}}_{\Uparrow}(x,y,t). \tag{G.27}$$

The reversible Wigner current is then defined in analogous fashion to the reversible classical current, while the irreversible Wigner current consists of all the remaining terms not in the reversible part. Although this definition is phenomenological, it makes sense on physical grounds since all contributions to the Wigner current not in the reversible component arise from irreversible processes. We thus define,

$$\overleftrightarrow{\boldsymbol{J}}_{\Uparrow}(x,y,t) = \begin{bmatrix} \omega_0 y \\ -\omega_0 x \end{bmatrix} W(x,y,t), \quad \vec{\boldsymbol{J}}_{\Uparrow}(x,y,t) = \boldsymbol{J}_{\Uparrow}(x,y,t) - \overleftrightarrow{\boldsymbol{J}}_{\Uparrow}(x,y,t). \tag{G.28}$$

As in the classical theory, we are interested in the steady-state Wigner current defined as,

$$\boldsymbol{j}_{\Uparrow}(x,y) = \lim_{t\to\infty} \boldsymbol{J}_{\Uparrow}(x,y,t). \tag{G.29}$$

The current $\boldsymbol{j}_{\Uparrow}(x,y)$ is thus defined by the steady-state Wigner function (A.33)–(A.35). From this, and noting that $K = \kappa_{\Uparrow}/\kappa_{\Downarrow}$, we can show explicitly that

$$\nabla \cdot \overleftrightarrow{\boldsymbol{j}}_{\Uparrow}(x,y) = 0, \quad \vec{\boldsymbol{j}}_{\Uparrow}(x,y) = \boldsymbol{0}. \tag{G.30}$$

where

$$\overleftrightarrow{\boldsymbol{j}}_{\Uparrow}(x,y) = \begin{bmatrix} \omega_0 y \\ -\omega_0 x \end{bmatrix} W_{\text{ss}}(x,y). \tag{G.31}$$

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
