# Peer review of "Quantum pure noise-induced transitions: A truly nonclassical limit cycle sensitive to number parity"

_SciPost Physics, doi:SciPost Phys. 15, 121 (2023)_

## Round 4 · Referee Report · Anonymous (Referee 1) · 2022-11-4

Strengths

1-interesting topic
2-detailed analysis
3-quantum and stochastic models

Weaknesses

1-appropriateness of concepts/definition (e.g. on limit cycles)
2-lack of proper focus in introduction (e.g. on P-bifurcations)
3-missing a clear statement about the novelty

Report

In this manuscript the authors analyse the stationary state of a bosonic mode with two-boson dissipation and gain from the perspective of noise-induced transitions. This is an interesting work with a broad spectrum of results and analytical insights. Moreover, while the results mainly concern the stationary state of the master equation, the authors also derive an equivalent stochastic formulation of the model that they use to physically interpret the properties of this stationary state. This thorough analysis is complemented by a comparison with a classical stochastic model described by a stochastic differential equation with an analogous mathematical structure, although it is not clear that such a model represents the classical counterpart of the quantum model. While we believe that this manuscript deserves publication, we think that before acceptance important issues should be addressed (see below). Also, depending on the answer, it will be clearer if this manuscript is more appropriate for SciPost or SciPost core.
Particularly important for us is that it is not clear that the observed phenomena can be thought of a generalization of classical limit cycles in the quantum domain, and thus that such a nomenclature might be misleading. This is because of the presence of a strong dynamical symmetry and thus the dependence of the observed phenomena on the initial conditions, something that contrasts with the limit-cycle scenarios.

Requested changes

-1- Conservative orbits vs. limit cycles: Limit cycles are generally defined as isolated closed trajectories in nonlinear classical systems (see for instance Strogatz book), being dynamical attractors. Different are closed conservative orbits: these are actually not unique (not isolated), with amplitude, e.g., depending on the initial condition. The case considered by the authors corresponds to a dissipative system but with conservative orbits. Indeed the stationary state (2), including the analysed closed orbits, depends on the initial condition since photon-number parity is a conserved quantity of the dynamical system. Moreover, the initial weight on the even/odd photon number sectors is used as a parameter in the bifurcation diagram, which further emphasizes the continuous variation of the observed closed orbits with the initial conditions. It does not seem therefore to be the quantum generalization of classical limit cycle.
Also, in the first section the authors introduce Hopf-bifurcations, while later what they report are P-bifurcations, a less known phenomenon, not related to the common instability scenario of the Hopf one.
Recommendation: use of correct nomenclature, consistently with the described features. Focus the introduction to the relevant context for the presented results or at least clarify in the introduction what this work is dealing with.

-2- Classical limit: The quantum system is fully characterized by the non-linear interaction with the environment (the unitary dynamics is just harmonic motion). The authors consider a classical limit that actually does not come from any mean field description (that would lead to a deterministic equation), nor to a semi-classical approximation (not obvious due to the presence of multiplicative noise), nor to a truncated Wigner approach. The choice to substitute quantum operators with classical fields is understandable to define a classical model (even if there are subtleties associated to the noise term), but it does not mean that this model would represent a macroscopic limit of the quantum microscopic model. Therefore statements like "We find that multiplicative quantum noise can induce a classically forbidden transition" are misleading and not properly justified.

-3-Novelty: Noise induced phenomena and transitions have been largely studied in the last decades, also in presence of quantum noise. Looking at the state of the art, the novelty of this work does not seem to be in "discovering" this phenomenon (in the quantum regime), nor in the considered multiplicative noise (common in quantum systems when departing from a quadratic Hamiltonian and/or linear losses), but instead in reporting, also with valuable analytical results, an insightful example with some interesting specific features.
Recommendations: That quantum noise can induce novel regimes does not seem the major breakthrough and the title could be more focused. As another example, at pg.2, there is a vague statement, about the motivation, to be improved: "In this work we show that pure noise-induced transitions can also occur in a microscopic system where quantum effects are essential." It is actually natural that, being the unitary part trivial, any relevant features will come from the form of noise. The main suggestion is to focus the introduction on the relevant context and specify the goal of the work more clearly.

-4-Multiplicative noise: the authors write "The most astonishing feature of multiplicative noise in nonlinear nonequilibrium systems is their ability to induce structured states, or “phases,” which without noise are completely absent." This is definitely not an exclusive feature of multiplicative noise. Additive noise does also induce regimes with no classical equivalent, as, for instance, convective instabilities in systems with spatial gradients.

  • validity: high
  • significance: good
  • originality: high
  • clarity: good
  • formatting: excellent
  • grammar: excellent

Author:  Changsuk Noh  on 2023-02-07  [id 3322]

(in reply to Report 1 on 2022-11-04)
Category:
reply to objection

In this manuscript the authors analyse the stationary state of a bosonic mode with two-boson dissipation and gain from the perspective of noise-induced transitions. This is an interesting work with a broad spectrum of results and analytical insights. Moreover, while the results mainly concern the stationary state of the master equation, the authors also derive an equivalent stochastic formulation of the model that they use to physically interpret the properties of this stationary state. This thorough analysis is complemented by a comparison with a classical stochastic model described by a stochastic differential equation with an analogous mathematical structure, although it is not clear that such a model represents the classical counterpart of the quantum model. While we believe that this manuscript deserves publication, we think that before acceptance important issues should be addressed (see below). Also, depending on the answer, it will be clearer if this manuscript is more appropriate for SciPost or SciPost core. Particularly important for us is that it is not clear that the observed phenomena can be thought of a generalization of classical limit cycles in the quantum domain, and thus that such a nomenclature might be misleading. This is because of the presence of a strong dynamical symmetry and thus the dependence of the observed phenomena on the initial conditions, something that contrasts with the limit-cycle scenarios.

We thank the referee for taking her/his time to review our manuscript. Our responses to the detailed comments can be found below. The corresponding changes in the main text are colored red (in the colored version of the manuscript we are attaching separately.) Note also that papers referred to in our replies here are either from the previous version of our manuscript, or appear at the end of the replies in a separate bibliography (such citations are denoted by [R1], [R2], and etc).

** Requested changes **

-1- Conservative orbits vs. limit cycles: Limit cycles are generally defined as isolated closed trajectories in nonlinear classical systems (see for instance Strogatz book), being dynamical attractors. Different are closed conservative orbits: these are actually not unique (not isolated), with amplitude, e.g., depending on the initial condition. The case considered by the authors corresponds to a dissipative system but with conservative orbits. Indeed the stationary state (2), including the analysed closed orbits, depends on the initial condition since photon-number parity is a conserved quantity of the dynamical system. Moreover, the initial weight on the even/odd photon number sectors is used as a parameter in the bifurcation diagram, which further emphasizes the continuous variation of the observed closed orbits with the initial conditions. It does not seem therefore to be the quantum generalization of classical limit cycle.

We thank the referee for pointing out this important issue. Our system does have a stable limit cycle, but this was not apparent due to our choice of the initial states. As the referee correctly notes, the system exhibits non-isolated orbits only if we restrict ourselves to initial states that are localized in phase space. For example, if we only initialize in coherent states, there is a one-to-one correspondence between the initial-state and steady-state amplitudes. In fact, using (44) and (94) from the Supplementary Material, we can see that 𝑛_{ss} = 2𝐾⁄(1 − 𝐾) + exp(−𝑛_0) sinh 𝑛_0, where 𝑛_0 and 𝑛_{ss} are respectively the initial and steady-state mean photon numbers. This seemingly suggests the absence of limit cycles.

However, this classical intuition breaks down for quantum systems since one can initialize in states whose Wigner functions are delocalized in phase space, e.g. arbitrary superpositions of Fock states. We emphasize that this is very different from initializing a classical system in an ensemble governed by some phase space (probability) distribution, which can also be delocalized. In the classical case, the evolution of the phase space distribution can always be regarded as the collection of independent trajectories of the point particles that make up the distribution. However, such interpretation is wrong in quantum dynamics, where the evolution of the Wigner function in general cannot be separated into independent evolutions (i.e. we cannot interpret the Wigner function of a quantum state as a probabilistic mixture of coherent-state Wigner functions, or any other ‘localized’ Wigner functions).

In fact, our closed orbits satisfy the key defining property of limit cycles, namely as attractors for a family of initial states [R1]. The Hilbert space of states is partitioned into two halves (even and odd parity sectors), with only the odd-parity part attracted to a unique closed orbit. For a fixed 𝐾, all initial states with the same initial photon-number parity (in Phases II and III) will tend towards the same closed orbit at steady state. This set of initial states consists of all coherent superpositions, or incoherent mixtures of states from these two sectors sharing the same photon-number parity, and in principle, spans over an infinite energy range and can be far from one another in the (quantum) phase space. Fundamentally, photon-number parity has no analogue in classical dynamics (as photons do not exist classically). The classical condition that states near an isolated orbit must be attracted to it, or repelled by it, is not necessary for a quantum limit cycle which is defined as a closed-orbit attractor in the quantum phase space. In fact, trajectories in the phase space is an ill-defined notion in quantum systems. In this regard, our proposed quantum limit cycle is a generalization of the classical limit cycle to quantum phase space, where the intrinsically quantum feature of parity conservation can play a central role, as in our case.

We have added a discussion on this point in the revised manuscript, Section III B.2 (p9).

Also, in the first section the authors introduce Hopf-bifurcations, while later what they report are P- bifurcations, a less known phenomenon, not related to the common instability scenario of the Hopf one. Recommendation: use of correct nomenclature, consistently with the described features. Focus the introduction to the relevant context for the presented results or at least clarify in the introduction what this work is dealing with.

We thank the referee for raising a possible point of confusion. To clarify the terminology, P-bifurcations is actually a method by which stochastic bifurcations are identified—phenomenologically by qualitative changes in the stationary probability density—as opposed to the more mathematical approach of dynamical bifurcations, or D-bifurcations for short (see Refs. [R2,R3] below and also Refs. [48—51] of the bibliography in the paper). Our use of P-bifurcations is also inline Ref. [44] of the paper, in which classical noise-induced transitions are also identified through the use of P-bifurcations. It is the method by which the physics literature identifies bifurcations and nonlinear features in quantum systems although physicists are not usually aware of this terminology. References [60, 61,114,118] are some examples that implicitly use P-bifurcations in a quantum system but without acknowledging it. We have tried to explain this on page 4 under Sec. III. A from which we quote our own text: To demonstrate noise-induced transitions in quantum systems we follow a similar approach as the classical theory [44], whereby the transition is characterized by the mode of the system’s steady-state probability density. This is formally known as phenomenological bifurcations, or P-bifurcations for short [110, 111]. Essentially the same idea applies for an open microscopic system, but to their quasiprobability distributions. The idea had been noted early on in quantum optics [112, 113]. Its use is now prevalent in physics (often without reference to P-bifurcations), such as in defining limit cycles near a Hopf bifurcation [60, 61], relaxation oscillations [114], amplitude and oscillation death [115–118], and Turing instabilities [119].

As for the bifurcation studied in this work, it is a stochastic Hopf bifurcation. In particular, our Hopf bifurcation is purely the result of quantum white noise—no quantum white noise, no Hopf bifurcation (and replacing the quantum white noise by classical white noise also fails to induce a stochastic Hopf bifurcation). Our references to P-bifurcations are meant to make explicit the method used to describe such a noise-induced Hopf bifurcation and the resulting limit cycle. We have now revised our manuscript to further clarify that P-bifucations is a method to describe nonlinear features and bifurcations, which should not be confused with the main focus of our paper, which is on noise-induced Hopf bifurcation/transition. The revision can be found at the end of the first paragraph on Section III A.

-2- Classical limit: The quantum system is fully characterized by the non-linear interaction with the environment (the unitary dynamics is just harmonic motion). The authors consider a classical limit that actually does not come from any mean field description (that would lead to a deterministic equation), nor to a semi-classical approximation (not obvious due to the presence of multiplicative noise), nor to a truncated Wigner approach. The choice to substitute quantum operators with classical fields is understandable to define a classical model (even if there are subtleties associated to the noise term), but it does not mean that this model would represent a macroscopic limit of the quantum microscopic model. Therefore statements like "We find that multiplicative quantum noise can induce a classically forbidden transition" are misleading and not properly justified.

Our quantum model does not have a semiclassical limit. We have tried to explain this on page 5 of the paper: Although our classical model is designed to mimick the quantum system, there are nevertheless some intrinsic differences. First, one often extracts the macroscopic limit of an open quantum system by performing a system- size expansion [97]. However, this method is only capable of extracting macroscopic systems with additive noise [61, 97]. Thus it cannot be employed to investigate whether multiplicative noise can induce limit cycles in a classical system. The system-size expansion therefore provides a sense in which our quantum oscillator lacks a classical limit. The usual method of system size expansion for obtaining semiclassical equations of motion fails for our model because multiplicative noise scales with the system size. Aside from such a mathematical understanding, a physical understanding can also be gained from Fig.1, which shows that multiplicative quantum noise arises from singly stimulated emissions (where one part of a two-photon emission is spontaneous, and one part stimulated)—a process that is entirely foreign to classical physics. This pictures shows that we cannot increase the system size (here the mean number of system photons), without also increasing the noise added to the system (here the number of spontaneous emissions).

However, we should not allow the lack of a formal semiclassical limit prevent us from considering any comparison to a similar classical stochastic system, albeit now defined by hand. Hence, for comparison purposes, in (9)—(11) we construct the closest classical approximation to the quantum stochastic dynamics. This is not to be misconstrued as the physical classical limit of a quantum system, as we have mentioned in the above quote from page 4 of the paper. It should also be noted that the truncated Wigner approach is a hand-waving way of extracting the semiclassical limit when it exists. The rigorous justification for it (and in fact also the technically correct way to do it) is via the system-size expansion as explained in Chap.5 of Ref. [97] in the paper.

However, we may still construct a classical stochastic model from the quantum Langevin equation by allowing the bosonic operators to commute and replacing them with c-numbers, which results in (9)— (11). This is an alternative way of defining the classical limit, and highlights the non-commutative nature of operators to be solely responsible for the noise-induced transitions in our paper, a genuine quantum effect, which are otherwise “classically forbidden.” Such a designation is not uncommon when discussing quantum effects. An elementary example is the quantum tunnelling of a particle into classically forbidden regions of a potential barrier, an effect which is once again fundamentally due to non-commuting operators in quantum mechanics.

We have added a few sentences to further clarify this point in the revised manuscript on p5, at the end of section II.

-3-Novelty: Noise induced phenomena and transitions have been largely studied in the last decades, also in presence of quantum noise. Looking at the state of the art, the novelty of this work does not seem to be in "discovering" this phenomenon (in the quantum regime), nor in the considered multiplicative noise (common in quantum systems when departing from a quadratic Hamiltonian and/or linear losses), but instead in reporting, also with valuable analytical results, an insightful example with some interesting specific features. Recommendations: That quantum noise can induce novel regimes does not seem the major breakthrough and the title could be more focused. As another example, at pg.2, there is a vague statement, about the motivation, to be improved: "In this work we show that pure noise-induced transitions can also occur in a microscopic system where quantum effects are essential." It is actually natural that, being the unitary part trivial, any relevant features will come from the form of noise. The main suggestion is to focus the introduction on the relevant context and specify the goal of the work more clearly.

The novelty of our work is a noise-induced transition which is both pure, and genuinely quantum. As we have explained in the literature review in Sec. I of our paper, previous work on state transitions in quantum systems with noise, such as quantum stochastic resonance [R4], synchronization bistability, or coherence resonance (respectively Refs. [63] and [65] of the paper) contains only transitions between states which are already present without noise. These transitions can be enabled simply by additive noise and a better term for them would be noise-activated processes. This leads us to the following two novelties of our noise-induced transition:

  1. The noise-induced transition is pure (see the treatise by Horsthemke and Lefever in Ref. [44] of our paper, especially Sec. 6.5): For this to happen, the noise affecting the system must induce a transition to a (stationary) state that is completely absent in the noiseless system. To achieve this one requires multiplicative noise. In our work, this new stationary state is the limit cycle, which would not exist at all if we switch off the multiplicative quantum white noise entering the system by setting the bath temperature to zero. To the best of our knowledge, our paper provides the first model of a pure noise-induced transition in a quantum system. References [62—86] of our paper, quantum or classical, all fall short of a pure noise-induced transition. For examples of classical systems that do qualify as a pure noise-induced transition see Refs. [54— 56] in our paper.

We have now made this point about Refs. [62—86] in our paper explicit in order to clarify the novelty of our work (see the sentences at the end of p2, the last paragraph of section I on p3, and a paragraph above Eq. (8)). We have also added two new figures, Figs. 1 and 2, for further clarification.

  1. The noise-induced transition is truly nonclassical: By this we mean, (i) The Wigner function representing the noise-induced limit cycle can become negative (phase III in Fig. 2 of the paper), which is a signature of a genuine quantum state. (ii) A classical stochastic model which best emulates the quantum stochastic model fails produce a limit cycle in any parameter regime. Therefore, even the limit cycles in phase II of Fig. 2 in our paper may be said to be nonclassical. (iii) Nonclassical steady states are formed in phases I to III. Using the Mandel Q-parameter as a nonclassicality witness, we have identified a subregion of nonclassicality in Fig. 2 which consists of states from all three phases. We have added this discussion in the revised manuscript, section III C.

    Note that we have also shown in Sec. V that exp(L⇑ 𝑡) induces a transition from a vacuum state (which is a classical state) to a nonclassical state only if noise is present, by which we mean 𝐾 > 0, and where L⇑ is given by (1) of the paper. This is interesting in that it shows that states in phase I of Fig. 2 (those along the ℘_+ = 1 line but excluding the point 𝐾 = 0) are in fact also nonclassical. It indirectly shows why the limit cycles of phase II are also nonclassical in the sense of (ii) above (see footnote 1 below).

The referee has pointed out that quantum systems with multiplicative noise is common (e.g. bosonic systems with multi-photon dissipation). However, in most of such works, the inherent multiplicative noise acts as a stochastic perturbation and does not lead to novel effects. One such example is the inhibition of perfect frequency locking in quantum synchronization [R5,R6], which is also expected in classical stochastic models. The novelty of our work lies in placing the focus on the quantum multiplicative noise which leads to ‘phase transitions’ which are not observable with the analogous classical multiplicative noise.

-4-Multiplicative noise: the authors write "The most astonishing feature of multiplicative noise in nonlinear nonequilibrium systems is their ability to induce structured states, or “phases,” which without noise are completely absent." This is definitely not an exclusive feature of multiplicative noise. Additive noise does also induce regimes with no classical equivalent, as, for instance, convective instabilities in systems with spatial gradients.

The statement referred to by the referee is the defining feature of pure noise-induced transitions in classical stochastic systems. It refers to what multiplicative noise can do in classical systems—namely induce states that are absent in a deterministic classical system (whereas additive noise cannot, see footnote 2). This is a widely accepted position within classical stochastic systems. We point to footnote 2 for a technical justification and simply quote a sample of the literature below.

• Ref. [56] from our paper, second paragraph in the Introduction on page 488 (note although they do not refer to this as a pure noise-induced transition it is clear that it is from reading their paper, and their Ref.15 refers to Horsthemke and Lefever): Notably, such residual stochasticity often manifests as a multiplicative, or state-dependent, noise at the collective level. In many cases, this can give rise to ‘finite- size noise-induced’ behaviour, where the probability of finding the system in a particular state is concentrated away from the deterministic (N⟶∞) fixed point(s).

• Ref. [49] from our paper, third paragraph in the Introduction on page 834: The shifts discussed here in systems which already deterministically exhibit a dynamical instability are one type of noise-induced transition treated by Horsthemke and Lefever, the other being the so-called “pure” noise-induced transition wherein bi- or multimodality is induced by noise in a system which deterministically has either no instability or a fewer number of instabilities.

The referee may have thus taken the quote out of context by comparing quantum to classical in his/her comment. What we have done in our work is to first investigate how multiplicative quantum white noise might cause a pure noise-induced transition, thereby extending the original idea of Horsthemke and Lefever to quantum theory. We then investigate if the pure noise-induced transition is genuinely quantum, which we find it is.

As a separate reply, we tried searching for the example mentioned by the referee and found some results on pattern formation due to convective instabilities and additive noise. This seems to be closely connected to the referee’s comments. One paper that appears to be representative of this line of research is Ref. [R7], which we comment on. To the best of our understanding, Ref. [R7] is not a pure noise-induced transition/effect. Their main result is that noise can sustain an optical pattern formed by the spatiotemporal variation of light intensity governed by a Ginzburg—Landau equation. By going into an appropriate parameter regime (the convectively unstable regime), a pattern can be produced by including additive noise. Reference [R7] considers an optical realization using a cavity containing a Kerr nonlinear medium and pumped by an external laser field. Physically, the regime of convectively instability is determined by the angle at which the pumping laser is launched into the cavity. This nonzero tilt of the pumping laser creates an advective (or spatial drift) term in the complex Ginzburg— Landau equation which is necessary for the convectively unstable regime. When the system is convectively unstable, local perturbations drift away from their point of origin in space and the same applies to a pattern. The key finding of Ref. [R7] is then, that by continuously injecting noise into the system (across space and time), one can regenerate the pattern at each point in space and time. The result is what the authors of Ref. [R7] refer to as a noise-sustained pattern. The noise-sustained pattern is not a pure noise-induced effect because the deterministic system already exhibits a pattern in its absolutely unstable regime. The noise does not induce a completely new state that is absent in the noiseless system (which here is the optical pattern), instead, noise only sustains it. In particular, the advective/spatial-drift term in the Ginzburg—Landau equation that is important to achieve convective instability relies on an entirely deterministic process, given by the launching angle of the external laser. In fact, no other parameter in the noise-added Ginzburg—Landau equation depend on the noise (other than the noise intensity itself). This is in contrast to our manuscript, which reports an effect that is purely the result of noise: If 𝜅⇑ = 0 in our model, then no limit cycle can be formed at all. This is because the temporal-drift term in our model (given by 𝜅⇑ \hat{a} 𝑑𝑡), which is necessary for the formation of a limit cycle, would vanish when 𝜅⇑ = 0. As explained in footnote 2 above, this is why pure noise-induced effects require multiplicative noise.

Reference [R7] is reminiscent of excitable systems and noise-sustained oscillations in purely temporal systems. In a deterministic excitable system such as the Fitzhugh—Nagumo model, there are two parameter regimes [R8]: One in which the system has only one stable fixed point, and another in which there is an attracting limit cycle. Noise-sustained oscillations may occur in this system if we tune the parameter such that the noiseless system enters its excitable regime for which the system has a fixed point, but is near the limit-cycle regime. In this case, the phase-space flow sufficiently far away from the fixed point is a big loop (due to the fact that it is approaching the limit-cycle regime). Then by including additive noise of sufficient strength, one can excite the system from its fixed point to such regions of phase space where it performs a loop before coming back to the fixed point again. For as long as the noise is present, the system may be continuously excited to perform such excursions/loops. Coherence resonance is then simply tuning the noise intensity so that these excursions in phase space occur at regular intervals. The result is referred to as oscillations (or sometimes a limit cycle) sustained by noise.

The two examples in Refs. [R7] and [R8] illustrate how additive-noise effects requires the deterministic system to contain nontrivial nonlinear effects prior to the addition of noise (a pattern for Ref. [R7] and a limit cycle for Ref. [R8]). In each case we first understand how the nonlinear phenomenon of interest is formed in the deterministic system and the various parameter regimes, then we tune the system away from the nonlinear phenomenon, but not too far away. Finally, including an appropriate amount of additive noise “restores” the nonlinear phenomenon. That such a procedure is possible relies on the noise being independent of the system, but this also means that added noise cannot introduce any extra terms (processes) into the system dynamics, which is why the deterministic system must exhibit a nontrivial nonlinear effect first. This is why Horsthemke and Lefever defines a pure noise-induced transition to be one taking place in a system where the nonlinear effect has to be completely absent in the corresponding noiseless system.

** Footnotes ** Footnote 1: It pays here to recall that a density operator is nonclassical if and only if the Glauber-Sudarshan P distribution is negative. This is the standard definition of nonclassicality in quantum optics and quantum information. The negativity of the Wigner function is only a sufficient condition for nonclassicality. Thus, states with a positive Wigner function may still be nonclassical because their Glauber-Sudarshan distribution might be negative. The single-mode squeezed state is one such example.

Footenote2: It is actually not too difficult to see why entirely new states absent in the deterministic system can be generated by multiplicative noise but not by additive noise: In defining a classical nonlinear system by 𝑥' = 𝑓(𝑥, 𝑦) and 𝑦' = 𝑔(𝑥, 𝑦), we capture all its nonlinear features, such as fixed points and limit cycles by 𝑓(𝑥, 𝑦) and 𝑔(𝑥, 𝑦). Since additive noise is independent of 𝑥 and 𝑦 (assumed to be white), adding them to 𝑓(𝑥, 𝑦) and 𝑔(𝑥, 𝑦) to define a new stochastic system simply perturbs the motion around the existing nonlinear features in a random fashion. That is, additive noise cannot create or destroy new features like changing the number of fixed points or creating a limit cycle when the noiseless system does not already have one. An example is a nonlinear system defined by a double-well potential, with two stable fixed points situated at the potential minima separated by a barrier. If we include additive noise in this system, and if the noise strength is sufficiently large to overcome the barrier separating the two wells, then the system may randomly jump between the two fixed points. This is often referred to as noise- induced bistability in the literature despite the “bistability” (i.e. the existence of two stable fixed points) being present in the noiseless system already. Multiplicative noise on the other hand introduces a white-noise term that depends on a nonlinear function of 𝑥 and 𝑦. This then has the ability to actually alter the nonlinear features such as creating new fixed points that did not exist before. In the context of bistability, we point to Ref. [55] in our paper as an example of a pure noise-induced transition (or rather “genuine” noise-induced bistability) where the multiplicative noise not only accounts for the stochastic jumps between two fixed points, but actually creates the fixed points (or creates a double-well potential so to speak, see also the final paragraph of Sec. V of our paper). This is fundamentally different to using a deterministic system containing two fixed points and then introducing additive noise to make the system jump between them. This ability of multiplicative noise to create brand new states can be mathematically seen from the Stratonovich-to-Ito conversion for stochastic differential equations. The conversion introduces nontrivial correction terms to the deterministic dynamics given by 𝑓(𝑥, 𝑦) and 𝑔(𝑥, 𝑦) for multiplicative noise. For additive noise, such correction terms vanish.

** References ** [R1] K. T. Alligood, T. D. Sauer, and J. A. Yorke, Chaos: An Introduction to Dynamical Systems, (Springer, 1996). [R2] L. Arnold, N. Sri Namachchivaya, and K. R. Schenk-Hoppé, Toward an understanding of stochastic Hopf bifurcation: A case study, Int. J. Bifurc. Chaos Appl. Sci. Eng. 6, 1947 (1996). [R3] K. R. Schenk-Hoppé, Stochastic Hopf bifurcation: An example, Int. J. Non-Linear Mech. 31, 685 (1996). [R4] T. Wagner, P. Talkner, J. C. Bayer, E. P. Rugeramigabo, P. Hänggi, and R. J. Haug, Quantum stochastic resonance in an a.c.-driven single-electron quantum dot, Nat. Phys. 15, 330 (2019). [R5] T. E. Lee and H. R. Sadeghpour, Quantum synchronization of quantum van der Pol oscillators with trapped ions, Phys. Rev. Lett. 111, 234101 (2013). [R6] S. Walter, A. Nunnekamp, and C. Bruder, Quantum synchronization of a driven self-sustained oscillator, Phys. Rev. Lett. 112, 094102 (2014). [R7] M. Santagiustina, P. Colet, M. S. Miguel, and D. Walgraef, Noise-sustained convective structures in nonlinear optics, Phys. Rev. Lett. 79, 3633 (1997). [R8] A. S. Pikovsky and J. Kurths, Coherence resonance in a noise-driven excitable system, Phys. Rev. Lett. 78, 775 (1997).

Attachment:

referee_report_1_y0iURoP.pdf

---

## Round 4 · Referee Report · Anonymous (Referee 2) · 2022-11-20

Strengths

1- Identification of a master equation which predicts limit cycles solely induced by the incoherent dynamics. 2- Detailed theoretical study (analytical and numerical) 3- Extensive review of existing literature.

Weaknesses

1- Some important assumptions need justification. 2- The interpretation of the results in terms of noise-induced resonances is questionable.

Report

This manuscript analyses the steady state and dynamical properties of a Lindbladian for a bosonic field which preserves the photon-number parity. The resulting steady-state is not unique and depends on the probability (below denoted by P), that the even photon number eigenspace is initially occupied. The steady state also depends the parameter K, namely, the ratio between heating processes (incoherent emission of two energy quanta) and cooling processes (incoherent absorption of two energy quanta). The properties of the steady state are analysed as a function of the probability P and of the parameter K in terms of (i) existence of a limit cycle - identified according to criteria that the authors introduce-, and (ii) negativity of the Wigner function. The results are compared with the predictions of a classical model introduced ad hoc.

Requested changes

Style.

The paper is generally written with care of details and with extensive discussions. The introduction of the paper provides a nice short review of noise-induced phenomena in quantum and classical physics. The paper also contains an extensive analytical analysis in the appendices. There are few typos (e.g., caption Fig. 1: “embbed” / after Eq. (23) in the appendices: “subtituting”). Suggestion: In Eq. (12). It would be useful to comment in the main text that x and y are the bosonic field’s quadratures. Remark: Sentence before Eq. (96): P_0 and P_1 are used in an argumentation but defined only later on.

Questions to the authors.

1) Thermal bath. In order to relate their study to noise-induced resonances, the authors refer to their model, Eq. (1), as a nonlinear damped oscillator coupled to a heat bath. In general, while Eq. (1) has an interesting theoretical justification on its own, I do not see how one can associate a temperature with it. In fact, Eq. (2) is generally not a thermal state. It is peculiar, that Eq. (2) does not even asymptotically reach a thermal state for K->1, which, according to the authors, shall be the case of very large temperatures. The authors shall provide an accurate justification of the use of the concept of temperature in their model (see also item 6 of this list)

2) Steady state I. Equation (2) does not contain off-diagonal elements between the even photon number and the odd photon number eigenspaces. The authors make reference to previous literature in order to motivate this ansatz. They also show that for K=0 this ansatz is not valid: In fact, in this limit there is a perfectly decoupled subspace spanned by the photon number states |0> and |1>. Since Eq. (2) shall hold for K>0, is the limit K->0 different from the case K=0? And in general, how shall one understand the limit of small K in the phase diagrams of Fig. 2 and 3? For instance: assume an initlal state that is a coherent superposition of 0 and 1 with equal probability: How do the off-diagonal elements between the two subspaces behave at long times for small K?

3) Steady state II. In connection with the previous item. Equation (109) in the appendices shall justify the form of Eq. (2). Is this formula also applicable to the steady state of the master equation with K=0?

4) Phase diagram I. Which physical observable shall distinguish Phase II from Phase III?

5) Phase diagram II. Some features of the phase diagram depend on K. However, one important result of this paper (Phase III) seems to solely depend on the initial state, and specifically on P. In view of this result, the claim of quantum noise-induced resonance is questionable.

6) Classical limit. The authors argue that Equation (9) shall provide a classical benchmark to their model, Eq. (1). The justification of this statement is not provided. The authors comment in the text about the difficulty of taking the classical limit of Eq. (1). As an alternative, they could consider the original model (emitters + photon field) and derive the Fokker-Planck equation in the semiclassical limit. In this case they would find a classical model with which they can consistently benchmark Eq. (1) (I suspect that they will have to take into account photon processes that break the symmetry of Eq. (1)). and also possibly identify the order of magnitude of K for which Eq. (1) is no longer valid.

  • validity: good
  • significance: good
  • originality: high
  • clarity: good
  • formatting: excellent
  • grammar: excellent

Author:  Changsuk Noh  on 2023-02-07  [id 3321]

(in reply to Report 2 on 2022-11-20)
Category:
answer to question
reply to objection

This manuscript analyses the steady state and dynamical properties of a Lindbladian for a bosonic field which preserves the photon-number parity. The resulting steady-state is not unique and depends on the probability (below denoted by P), that the even photon number eigenspace is initially occupied. The steady state also depends the parameter K, namely, the ratio between heating processes (incoherent emission of two energy quanta) and cooling processes (incoherent absorption of two energy quanta). The properties of the steady state are analysed as a function of the probability P and of the parameter K in terms of (i) existence of a limit cycle - identified according to criteria that the authors introduce-, and (ii) negativity of the Wigner function. The results are compared with the predictions of a classical model introduced ad hoc.

We thank the referee for taking her/his time to review our manuscript. Our responses to detailed comments can be found below. The corresponding changes in the main text are colored blue (in the colored version of the manuscript we are attaching separately.) Note also that papers referred to in our replies here are either from the previous version of our manuscript, or appear at the end of the replies in a separate bibliography (such citations are denoted by [R1], [R2], and etc).

** Requested changes ** Style.

The paper is generally written with care of details and with extensive discussions. The introduction of the paper provides a nice short review of noise-induced phenomena in quantum and classical physics. The paper also contains an extensive analytical analysis in the appendices. There are few typos (e.g., caption Fig. 1: “embbed” / after Eq. (23) in the appendices: “subtituting”). Suggestion: In Eq. (12). It would be useful to comment in the main text that x and y are the bosonic field’s quadratures. Remark: Sentence before Eq. (96): P_0 and P_1 are used in an argumentation but defined only later on.

We thank the referee for carefully reading the manuscript. We have fixed the typos in the revised manuscript.

Questions to the authors. 1) Thermal bath. In order to relate their study to noise-induced resonances, the authors refer to their model, Eq. (1), as a nonlinear damped oscillator coupled to a heat bath. In general, while Eq. (1) has an interesting theoretical justification on its own, I do not see how one can associate a temperature with it. In fact, Eq. (2) is generally not a thermal state. It is peculiar, that Eq. (2) does not even asymptotically reach a thermal state for K->1, which, according to the authors, shall be the case of very large temperatures. The authors shall provide an accurate justification of the use of the concept of temperature in their model (see also item 6 of this list)

We thank the referee from raising a possible point of confusion. It is important to mention that we only refer to “temperature” as the temperature of the atomic bath which is traced out in master equation (1) (see the opening paragraph of Sec. II of our paper). The atomic ensemble in the bath is thus in a thermal state at temperature 𝑇 [see (27) in the Supplementary Material of Ref. [57] from our paper for a detailed discussion]. Therefore, the oscillator is not thermalized to temperature 𝑇, as pointed out by the referee. However, the parameter 𝐾 increases with the temperature (and hence thermal noise) of the bath, so 𝐾 ⟶ 1 for the oscillator is equivalent to 𝑇 ⟶ ∞ for the atomic bath.

We have also revised our references to “the temperature” in the Introduction as being the temperature of the bath so that this point is made clear at least twice (once in Sec. I and once more in Sec. II).

2) Steady state I. Equation (2) does not contain off-diagonal elements between the even photon number and the odd photon number eigenspaces. The authors make reference to previous literature in order to motivate this ansatz. They also show that for K=0 this ansatz is not valid: In fact, in this limit there is a perfectly decoupled subspace spanned by the photon number states |0> and |1>. Since Eq. (2) shall hold for K>0, is the limit K->0 different from the case K=0? And in general, how shall one understand the limit of small K in the phase diagrams of Fig. 2 and 3? For instance: assume an initlal state that is a coherent superposition of 0 and 1 with equal probability: How do the off-diagonal elements between the two subspaces behave at long times for small K?

Yes, the 𝐾 = 0 case is qualitatively different from the 𝐾 = 0 case. For 𝐾 = 0 , one has an extra conserved quantity for the off-diagonal coherence on top of parity conservation [see (3.14) of Ref. [131] in our paper for details], while the 𝐾 > 0 case only has parity conservation. Hence, the phases plotted on the left-edge of Fig. 2 in the old manuscript is strictly speaking for the 𝐾 ⟶ 0^+ limit. This marks a thermal noise-induced transition from 𝐾 = 0 to the various phases shown in Figs. 2 and 3 of the old manuscript depending on the initial parity.

As per the referee’s question, if we initialize in the coherent superposition of the even and odd parity states, the off-diagonal coherence will decay to zero on a timescale of 𝐾^{-1}. Hence, the parity is conserved in the steady state only if 𝐾 = 0.

We have added a few sentences to clarify this point in the revised text. See the end of the first paragraph on p6.

3) Steady state II. In connection with the previous item. Equation (109) in the appendices shall justify the form of Eq. (2). Is this formula also applicable to the steady state of the master equation with K=0?

Yes, (109) from our Supplementary Material is applicable even if 𝐾 = 0, but in that case there will be an additional conserved quantity for the coherence as mentioned in the previous answer, so we have D = 3 instead.

4) Phase diagram I. Which physical observable shall distinguish Phase II from Phase III?

Phases II and III can be distinguished by measuring the parity of the state \hat{\Pi} = (−1)^{a^\dagger a} , which is related to the Wigner negativity using 𝑊(0) = 2Tr[\hat{\Pi}\rho]⁄π. The expectation value of parity can be obtained in experiments via non-demolition measurements [R1].

5) Phase diagram II. Some features of the phase diagram depend on K. However, one important result of this paper (Phase III) seems to solely depend on the initial state, and specifically on P. In view of this result, the claim of quantum noise-induced resonance is questionable.

Our quantum noise-induced transitions refer to (i) the formation of various phases I to III from 𝐾 = 0 to 𝐾 > 0 as discussed in a previous answer; and (ii) the stochastic Hopf bifurcation between phases I and II which occurs for a critical 𝐾. The justification for “quantum noise-induced” transitions comes from the comparison with the analogous classical noisy model which shows no such behaviour in all parameter regimes. We have added a few sentences on the second paragraph of p2 to explain this.

As the referee has noted, once the state is in Phase III, no amount of thermal noise can remove the Wigner negativity. This is equivalent to the conservation of parity and is not related to the stochastic Hopf bifurcation.

6) Classical limit. The authors argue that Equation (9) shall provide a classical benchmark to their model, Eq. (1). The justification of this statement is not provided. The authors comment in the text about the difficulty of taking the classical limit of Eq. (1). As an alternative, they could consider the original model (emitters + photon field) and derive the Fokker-Planck equation in the semiclassical limit. In this case they would find a classical model with which they can consistently benchmark Eq. (1) (I suspect that they will have to take into account photon processes that break the symmetry of Eq. (1)). and also possibly identify the order of magnitude of K for which Eq. (1) is no longer valid.

As mentioned in the page 4 of the main text “Although our classical model ... completely deterministic”, the usual method of system size expansion for obtaining semiclassical Fokker-Planck equation of motion fails for our model due to the presence of multiplicative noise which scales with the system size. Other equivalent ways of obtaining semiclassical limits like truncated Wigner approximation and mean-field approximation will fail for the same reason. Fundamentally, our model lacks a proper classical limit unlike quantum models with additive noise. [See the added sentences in red on p5 (also a response to referee 1), at the end of section II.]

Hence, for comparison purposes, we construct the closest classical approximation to the noisy dynamics given in (10). The classical noise model is constructed from the quantum Langevin equation by allowing the bosonic operators to commute and replacing them with c-numbers. This is an alternative method to find the semiclassical limit. This is not to be misconstrued as the physical classical limit of a quantum system, as we have mentioned in page 4. Our approach amounts to allowing the bosonic operators to commute and replacing them with c-numbers. Therefore, the differences in behaviour between the quantum model (1) and the analogous classical model (10) can be attributed to the non-commuting nature of quantum operators, which is a genuine quantum effect.

We remark that for additive noise, this procedure is equivalent to a semiclassical approximation but this is not true for multiplicative noise in general (see the second paragraph in Sec. 4 on page 5 of Ref. [R2], also cited as Ref. [63] in our paper). We also refer to our reply to the second point raised in the first referee report. References

[R1] L. Sun, A. Petrenko, Z. Leghtas, B. Vlastakis, G. Kirchmair, K. M. Sliwa, A. Narla, M. Hatridge, S. Shankar, J. Blumoff, L. Frunzio, M. Mirrahimi, M. H. Devoret, and R. J. Schoelkopf, Tracking photon jumps with repeated quantum non-demolition parity measurements, Nature 511, 444 (2014). [R2] T. Weiss, A. Kronwald, and F. Marquardt, Noise-induced transitions in optomechanical synchronization, New J. Phys. 18, 013043 (2016).

Attachment:

referee_report_2_zZ2rZf6.pdf

Author:  Changsuk Noh  on 2023-02-07  [id 3316]

(in reply to Report 2 on 2022-11-20)

Please find the attached reply.

Attachment:

referee_report_1.pdf

Author:  Changsuk Noh  on 2023-02-07  [id 3315]

(in reply to Report 2 on 2022-11-20)

Please find the attached reply.

Attachment:

referee_report_2.pdf

---

## Round 6 · Author Response

We would like to thank the editor for handling our manuscript and the referees for their comments. Please find below our replies to the comments.

Anonymous report 1: The authors have revised the paper and addressed most of my concerns. The presentation has substantially improved. The comparison with the “classical” model remains artificial. The absence of a rigorous classical limit of the model the authors analyse is not clearly emphasized. The corresponding claims in the paper keep on being confusing and shall be revised (see “We find that multiplicative quantum noise can induce a classically forbidden transition” in the abstract, and similar claims across the paper).

Our reply: We thank the referee for his/her report. It is unclear to us why the referee is emphasizing the lack of a classical limit in our quantum model as this is in fact a highly novel feature. We have now incorporated the referee’s comments in the new edition of our paper. There are now three locations in the manuscript where we state that our quantum model has no classical limit:

  1. We would like to point out that, on the previous version of our paper, we had already explained explicitly the absence of a classical limit as defined by the system-size expansion where the classical model was first introduced (see the first paragraph on the right hand column on page 5).

  2. We have now stated the absence of a classical limit in the paper abstract by adding a sentence "without a classical limit".

  3. Finally, the lack of a classical limit is brought up again in the conclusion of our paper (the last paragraph), where we have also reminded the reader why our quantum-classical comparison is worthwhile even though the quantum system has no classical limit.

On a separate point, we have voluntarily included a new reference as part of our literature review (Ref. [87] in the new bibliography).

Anonymous report 2 The authors have provided a complete answer to the main questions raised in my first report and also significantly changed the manuscript (including illustrations, definitions, and clarifications). The nature of the process and the novelty of the reported limit cycle is now clear. Also, the connection with the classical process has been better justified. The analysis of this problem is non-trivial and the results are original. The added material does improve the readability of this work making it accessible to a broader audience. Furthermore, the detailed author's answers will be useful. After considering the author's answer, I better appreciate the novelty of the work. Even if the research community interested in quantum dynamics of complex systems is not the broadest, this pure noise-induced limit cycle is actually a nice quantum effect and in my opinion, this work deserves publication in SciPost.

Our reply: We thank the reviewer for taking the time to examine our previous reply. We have nothing to add except for including Ref. [87] in our literature review (the results of which are actually classical, see red text on page 2).

---

## Round 6 · List of Changes

1. Added a sentence "...without a classical limit" in the abstract.
  2. Added a sentence "The effect of multiplicative noise on a bistable..." on p3 to add a new reference [87].
  3. Added a paragraph to the conclusion.

---

## Editorial Decision

published